# Self-Supervised Reinforcement Learning that Transfers using Random Features

**Boyuan Chen**[*]
Massachusetts Institute of Technology
Boston, MA 02139
boyuanc@mit.edu

**Chuning Zhu**[*]
University of Washington
Seattle, WA 98105
zchuning@cs.washington.edu

**Pulkit Agrawal**
Massachusetts Institute of Technology
Boston, MA 02139
pulkitag@mit.edu

**Kaiqing Zhang**[†]
University of Maryland
College Park, MD 20742
kaiqing@umd.edu

**Abhishek Gupta**[†]
University of Washington
Seattle, WA 98105
abhgupta@cs.washington.edu

## Abstract

Model-free reinforcement learning algorithms have exhibited great potential in solving single-task sequential decision-making problems with high-dimensional observations and long horizons, but are known to be hard to *generalize* across tasks. Model-based RL, on the other hand, learns task-agnostic models of the world that naturally enables transfer across different reward functions, but struggles to scale to complex environments due to the compounding error. To get the best of both worlds, we propose a self-supervised reinforcement learning method that enables the transfer of behaviors across tasks with different rewards, while circumventing the challenges of model-based RL. In particular, we show self-supervised pretraining of model-free reinforcement learning with a number of *random features* as rewards allows *implicit* modeling of long-horizon environment dynamics. Then, planning techniques like model-predictive control using these implicit models enable fast adaptation to problems with new reward functions. Our method is self-supervised in that it can be trained on offline datasets *without* reward labels, but can then be quickly deployed on new tasks. We validate that our proposed method enables transfer across tasks on a variety of manipulation and locomotion domains in simulation, opening the door to generalist decision-making agents.

## 1 Introduction

As in most machine learning problems, the ultimate goal of building deployable sequential decision-making agents via reinforcement learning (RL) is its broad *generalization* across tasks in the real world. While reinforcement learning algorithms have been shown to successfully synthesize complex behavior in *single-task* sequential decision-making problems [31, 36, 45], their performance as *generalist agents* across tasks has been less convincing [9]. In this work, we focus on the problem of learning generalist agents that are able to transfer across problems where the environment dynamics are shared, but the reward function is changing. This problem setting is reflective of scenarios

---

[*]Equal contribution.
[†]Equal advising.

37th Conference on Neural Information Processing Systems (NeurIPS 2023).

that may be encountered in real-world settings such as robotics. For instance, in tabletop robotic manipulation, different tasks like pulling an object, pushing an object, picking it up, and pushing to different locations, all share the same transition dynamics, but involve different reward functions. We hence ask the question – *Can we reuse information across tasks in a way that scales to high dimensional, longer horizon problems?*

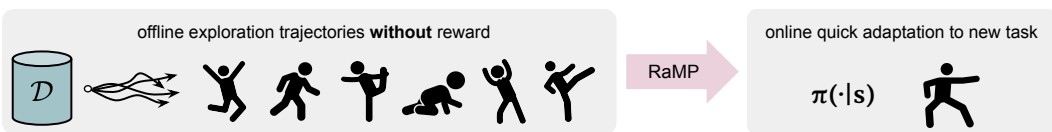

Figure 1: `RaMP` is a self-supervised reinforcement learning algorithm that pre-training on exploration trajectories without (or with unknown) reward. It then quickly adapts to new tasks through interaction with the environment.

A natural possibility to tackle this problem is direct policy search [54, 44]. Typical policy search algorithms can achieve good performance on a single task. However, the policy is optimized for a particular reward and may be highly suboptimal in new scenarios. Other model-free RL algorithms like actor-critic methods [23, 15, 13] or Q-learning [51, 31] may exacerbate this issue, with learned Q-functions entangling dynamics, rewards, and policies. For new scenarios, an ideal algorithm should be able to disentangle and retain the elements of shared dynamics, allowing for easy substitution of rewards without having to repeat significant computations. While multi-task policy search techniques like goal-conditioned RL address this issue to some extent, they are restricted to particular tasks like goal-reaching and usually require prior knowledge about the tasks one may encounter in testing.

Model-based RL arises as a natural fit for disentangling dynamics and rewards [33, 10, 17, 19, 20]. These algorithms directly learn a model of transition dynamics and leverage the learned model to plan control actions [35, 17, 10, 8]. In particular, the models can be used to *re-plan* behaviors for new rewards. However, these learned dynamics models are brittle and suffer from compounding error [25, 3] due to their autoregressive nature. States predicted by the model are iteratively used as inputs for subsequent predictions, leading to compounding approximation errors during planning [25]. This makes it challenging to apply them to problems with long horizons and high dimensions.

In this work, we thus ask – *Can we build RL algorithms that disentangle dynamics, rewards, and policies for transfer across problems, while retaining the ability to solve problems with high dimensional observations and long horizons?* To this end, we propose a self-supervised RL algorithm that can leverage collected data to *implicitly* model transition dynamics without ever having to *generate* future states, and then use this to quickly transfer to a variety of different new tasks with varying reward functions that may be encountered at test time.

Our method relies on the following key insight: to prevent tying specific reward functions and policies to the RL algorithm, one can instead model the long-term evolution of randomly chosen basis functions of all possible reward functions [6]. Such basis functions can be generated using random features [39–41]. The accumulation of such basis functions over time, which we refer to as a Q-basis, is task-agnostic [48], and *implicitly* models the transition dynamics of the problem. We show that with a *large enough* number of random features and model-free RL, the value function for *any* reward function that may be encountered in testing could be estimated, via a simple linear combination of these Q-basis functions. This is important since it allows us to obtain the benefits of transfer, without explicitly learning a dynamics model that may suffer from compounding error. Moreover, with randomly chosen basis functions that are agnostic of downstream tasks, this is not restricted to just a certain distribution of tasks, as in most multi-task or goal-conditioned RL [49, 2]. Finally, we note that the random features (as well as the corresponding Q-basis functions) can be generated using datasets that are not necessarily *labeled* with reward signals. Indeed, these random-feature-based pseudo rewards can be viewed as self-supervised signals in training, making the approaches based on them naturally compatible with large (task-agnostic) offline datasets.

Based on the aforementioned insights, we propose a new algorithm *Random Features for Model-Free Planning* (`RaMP`) that allows us to leverage unlabelled offline datasets to learn reward-agnostic Q-bases. These can be used to estimate the test-time value functions for new reward functions using linear regression, enabling quick adaptation to new tasks under the same shared transition dynamics. We show the efficacy of this method on a number of tasks for robotic manipulation and locomotion in simulation, and highlight how `RaMP` provides a more general paradigm than typical generalizations of model-based or model-free RL.

## 1.1 Related Work

Model-based RL is naturally suited for transfer across tasks with shared dynamics, by explicitly learning a model of the transition dynamics, which is disentangled with the reward functions for each task [17, 35, 10, 16, 8, 47, 27]. These models are typically learned via supervised learning on one-step transitions and then used to extract control actions via planning [28, 42] or trajectory optimization [46, 59, 38]. The key challenge in scaling lies in the fact that they sequentially feed model predictions back into the model for sampling [50, 3, 25]. This can often lead to compounding errors [25, 3, 55], which grow with the horizon length unfavorably. In contrast, our work does not require autoregressive sampling, making it easier to scale to longer horizons and higher dimensions.

On the other hand, model-free RL avoids the challenge of compounding error by directly modeling either policies or Q-values without autoregressive generation [54, 43, 44, 30, 31, 15]. However, these approaches entangle rewards, dynamics, and policies, making them hard to transfer. While attempts have been made to build model-free methods that generalize across rewards, such as goal-conditioned value functions [21, 1, 37, 14] or multi-task policies [22, 18], they only apply to restricted classes of reward functions and particular training distributions. Our work aims to obtain the best of both worlds (model-based and model-free RL), learning some representation of dynamics independent of rewards and policies, while using a model-free algorithm for learning.

Our notion of long-term dynamics is connected to the notion of state-action occupancy measure [34, 58], often used for off-policy evaluation and importance sampling methods in RL. These methods often try to directly estimate either densities or density ratios [20, 34, 58]. Our work simply learns the long-term accumulation of random features, without requiring any notion of normalized densities. Perhaps most closely related work to ours is the framework of *successor features*, that considers transfer from a fixed set of source tasks to new target tasks [5, 26, 57, 11]. Like our work, the successor features framework leverages the linearity of rewards to disentangle dynamics from rewards using model-free RL. However, transfer using successor features is dependent on choosing (or learning) the right featurization and forces an implicit dependence on the policy. Our work leverages random features and multi-step Q-functions to alleviate the transfer performance of successor features.

## 2 Preliminaries

**Model.** We consider the standard Markov decision process (MDP) as characterized by a tuple $\mathcal{M} = (\mathcal{S}, \mathcal{A}, \mathcal{T}, R, \gamma, \mu)$, with state space $\mathcal{S}$, action space $\mathcal{A}$, transition dynamics $\mathcal{T} : \mathcal{S} \times \mathcal{A} \to \Delta(\mathcal{S})$, reward function $R : \mathcal{S} \times \mathcal{A} \to \Delta([-R_{\max}, R_{\max}])$, discount factor $\gamma \in [0, 1)$, and initial state distribution $\mu \in \Delta(\mathcal{S})$. The goal is to learn a policy $\pi : \mathcal{S} \to \Delta(\mathcal{A})$, such that it maximizes the expected discounted accumulated rewards, i.e., solves $\max_\pi \mathbb{E}_\pi \left[ \sum_{h=1}^\infty \gamma^{h-1} r_h \right]$ with $r_h :=$ $r(s_h, a_h) \sim R_{s_h, a_h} = \Pr(\cdot \,|\, s_h, a_h)$. Hereafter, we will refer to an MDP and a *task* interchangeably. Note that in our problem settings, different MDPs will always share transition dynamics $\mathcal{T}$, but will have varying reward functions $R$.

**Q-function Estimation.** Given an MDP $\mathcal{M}$, one can define the state-action $Q$-value function under any policy $\pi$ as $Q_\pi(s, a) := \mathbb{E}_{\substack{a_h \sim \pi(\cdot \,|\, s_h) \\ s_{h+1} \sim \mathcal{T}(\cdot \,|\, s_h, a_h)}} \left[ \sum_{h=1}^\infty \gamma^{h-1} r_h \,\Big|\, s_1 = s, a_1 = a \right]$ which denotes the expected accumulated reward under policy $\pi$, when starting from state-action pair $(s, a)$. By definition, this $Q$-function is inherently tied to the particular reward function $R$ and the policy $\pi$, making it challenging to transfer for a new reward or policy. Similarly, one can also define the *multi-step ($\tau$-step) Q-function* $Q_\pi(s, \widetilde{a}_1, \widetilde{a}_2, \cdots, \widetilde{a}_\tau) = \mathbb{E}_{\substack{a_{\tau+h} \sim \pi(\cdot \,|\, s_{\tau+h}) \\ s_{h+1} \sim \mathcal{T}(\cdot \,|\, s_h, a_h)}} \left[ \sum_{h=1}^\infty \gamma^{h-1} r_h \,\Big|\, s_1 = s, a_1 = \widetilde{a}_1, a_2 = \widetilde{a}_2, \cdots, a_\tau = \widetilde{a}_\tau \right]$.

One can estimate the $\tau$-step $Q_\pi$ by Monte-Carlo sampling of the trajectories under $\pi$, i.e., by solving

$$\min_{\widehat{Q} \in \mathcal{Q}} \quad \frac{1}{N} \sum_{j=1}^N \left\| \widehat{Q}(s, \widetilde{a}_1^j, \widetilde{a}_2^j, \cdots, \widetilde{a}_\tau^j) - \frac{1}{M} \sum_{m=1}^M \sum_{h=1}^\infty \gamma^{h-1} r_h^{m,j} \right\|_2^2, \tag{2.1}$$

where $\mathcal{Q}$ is some function class for $Q$-value estimation, which in practice is some parametric function class, e.g., neural networks; $r_h^{m,j} \sim R_{s_h^{m,j}, a_h^{m,j}}$ and $(s_h^{m,j}, a_h^{m,j})$ come from $MN$ trajectories that

are generated by $N$ action sequences $\{(\widetilde{a}_1^j, \widetilde{a}_2^j, \cdots, \widetilde{a}_\tau^j)\}_{j=1}^N$ and $M$ trajectories following policy $\pi$ after each action sequence. A large body of work considers finding this $Q$-function using dynamic programming, but for the sake of simplicity, this work will only consider Monte-Carlo estimation.

In practice, the infinite-horizon estimator in (2.1) can be hard to obtain. We hence use a finite-horizon approximation of $Q_\pi$ (of length $H$), denoted by $Q_\pi^H$, in learning. We will treat this finite-horizon formula in the main paper for practical simplicity. We also provide a method to deal with the infinite-horizon case directly by bootstrapping the value function estimate. We defer the details to Appendix B.1. Note that if one chooses $H = \tau$, then the $\tau$-step $Q$-function defined above becomes

$$Q_\pi^H(s, \widetilde{a}_1, \widetilde{a}_2, \cdots, \widetilde{a}_H) := \mathbb{E}_{s_{h+1} \sim \mathcal{T}(\cdot \,|\, s_h, a_h)} \Big[ \sum_{h=1}^H \gamma^{h-1} r_h \,|\, s_1 = s, a_1 = \widetilde{a}_1, \cdots, a_H = \widetilde{a}_H \Big].$$

Note that in this case, the $Q$-function is irrelevant of the policy $\pi$, denoted by $Q^H$, and is just the expected accumulated reward executing the action sequence $(\widetilde{a}_1, \widetilde{a}_2, \cdots, \widetilde{a}_H)$ starting from the state $s$. This $Q$-function can be used to score how "good" a sequence of actions will be, which in turn can be used for planning.

**Problem setup.** We illustrate our problem setup in Figure 1. Consider a transfer learning scenario, where we assume access to an offline dataset consisting of several episodes $\mathcal{D} = \{(s_h^m, a_h^m, s_{h+1}^m)\}_{h \in [H], m \in [M]}$. Here $H$ is the length of the trajectories, which is large enough, e.g., of order $\mathcal{O}(1/(1-\gamma))$ to approximate the infinite-horizon setting; $M$ is the total number of trajectories. This dataset assumes that all transitions are collected under the same transition dynamics $\mathcal{T}$, but otherwise does not require the labels, i.e., rewards, and may even come from different behavior policies. This is reminiscent of the offline RL problem statement, but offline RL datasets are typically *labeled* with rewards and are *task-specific*. In contrast, in this work, the pre-collected dataset is used to quickly learn policies for *any* downstream reward, rather than for one specific reward function. The goal is to make the best use of the dataset $\mathcal{D}$, and generalize the learned experience to improve the performance on a new task $\mathcal{M}$, with the same transition dynamics $\mathcal{T}$ but an arbitrary reward function $R$. Note that unlike some related work [5, 4], we make *no* assumption on the reward functions of the MDPs that generate $\mathcal{D}$. The goal of the learning problem is to *pre-train* on the offline dataset such that we can enable fast (even zero-shot) adaptation to arbitrary new reward functions encountered at test time.

## 3  `RaMP`: Learning Implicit Models for Cross-Reward Transfer with Self-Supervised Model-Free RL

We now introduce our algorithm, Random Features for Model-Free Planning (`RaMP`), to solve the problem described in Section 2 – learning a model of long-term dynamics that enables transfer to tasks labeled with arbitrary new rewards, while making use of the advantages of model-free RL. Clearly, the problem statement we wish to solve requires us to (1) be able to solve tasks with long horizons and high-dimensional observations and (2) alleviate policy and reward dependence to be able to transfer computation across tasks. The success of model-free RL in circumventing compounding error from autoregressive generation raises the natural question: *Is there a model-free approach that can mitigate the challenges of compounding error and can transfer across tasks painlessly?* We answer this in the affirmative in the following section.

### 3.1  Key Idea: Implicit Model Learning with Random Feature Cumulants and Model-Free RL

The key insight we advocate is that we can avoid task dependence if we directly model the long-term accumulation of many random functions of states and actions (treating them as the rewards), instead of modeling the long-term accumulation of one specific reward as in typical Q-value estimation. Since these random functions of the state and actions are task-agnostic and uninformed of any specific reward function, they simply capture information about the transition dynamics of the environment. However, they do so without actually requiring autoregressive generative modeling, as is commonplace in model-based RL. Doing so can effectively disentangle transition dynamics from reward, and potentially allow for transfer across tasks, while still being able to make use of model-free RL methods. Each long-term accumulation of random features is referred to as an element of a "random" Q-basis, and can be learned with simple modifications to typical model-free RL algorithms. The key is to replace the Q-estimation of a single *task-specific* reward function with estimating the Q-functions of a set of *task-agnostic* random functions of the state and actions as a random Q-basis.

At *training time*, the offline dataset $\mathcal{D}$ can be used to learn a set of "random" Q-basis functions for different random functions of the state and action. This effectively forms an "implicit model", as it carries information about how the dynamics propagate, without being tied to any particular reward function. At *test time*, given a new reward function, we can recombine Q-basis functions to effectively approximate the true reward-specific Q-function under any policy. This inferred Q-function can then be used for planning for the new task. As we will show in Section 3.4, this recombination can actually be done by a linear combination of Q-basis functions for a sufficiently rich class of random features, reducing the problem of test-time adaptation for new rewards to a simple linear regression problem. We detail the two phases of our approach in the following subsections.

Figure 2: RaMP: Depiction of our proposed method for transferring behavior across tasks by leveraging model-free learning of random features. At training time, Q-basis functions are trained on accumulated random features. At test time, adaptation is performed by solving linear regression and recombining basis functions, followed by online planning with MPC.

## 3.2 Offline Training: Self-Supervised Learning Random Q-functions

Without any prior knowledge about the downstream test-time rewards, the best that an agent can do is to model the evolution of the state (i.e., model system dynamics). The key insight we make is that the evolution of state can be implicitly represented by simply modeling the long-term accumulation of *random* features of state to obtain a *set* of Q-basis functions. Such a generation of random features is fully self-supervised. These functions can then be combined to infer the task-specific Q-function.

Given the lack of assumptions about the downstream task, the random features being modeled must be expressive and universal in their coverage. As suggested in [39–41], random features can be powerful in that most nonlinear functions can be represented as linear combinations of random features effectively. As we will show in Section 3.4, modeling the long-term accumulation of random features allows us express the value function for *any* reward as a **linear** combination of these accumulations of random features. In particular, in our work, we assume that the random features are represented as the output of $K$ neural networks $\phi(\cdot, \cdot; \theta_k) : \mathcal{S} \times \mathcal{A} \to \mathbb{R}$ with weights $\theta_k \in \mathbb{R}^d$ and $k \in [K]$, where $\theta_k$ are *randomly* sampled i.i.d. from some distribution $p$. Sampling $K$ such weights $\theta_k$ with $k \in [K]$ yields a vector of scalar functions $[\phi(\cdot, \cdot; \theta_k)]_{k \in [K]} \in \mathbb{R}^K$ for any $(s, a)$, which can be used as random features. To model the long-term accumulation of each of these random features, we note that they can be treated as reward functions (as can any function of state and action [48]) in model-free RL, and standard model-free policy evaluation to learn Q-functions can be reused to learn a set of $K$ Q-basis functions, with each of them corresponding to the Q-value of a random feature.

We note that done naively, this definition of a Q-basis function is tied to a particular policy $\pi$ that generates the trajectory. However, to transfer behavior one needs to predict the accumulated random features under new sequences of actions, as policy search for the new task is likely to involve evaluating a policy that is not the same as the training policy $\pi$. To allow the modeling of accumulated features that is independent of particular policies, we propose to learn *multi-step* Q-basis functions for each of the random features (as discussed in Section 2), which is explicitly dependent on an input sequence of actions $(\tilde{a}_1, \tilde{a}_2, \cdots, \tilde{a}_\tau)$. This can be used to search for optimal actions in new tasks.

To actually learn these Q-basis functions (one for each random feature), we use Monte-Carlo methods for simplicity. We generate a new dataset $\mathcal{D}_\phi$ from $\mathcal{D}$, with $\mathcal{D}_\phi = \{((s_1^m, a_{1:H}^m), \sum_{h \in [H]} \gamma^{h-1} \phi(s_h^m, a_h^m; \theta_k))\}_{m \in [M], k \in [K]}$. Here we use

$\sum_{h \in [H]} \gamma^{h-1} \phi(s_h^m, a_h^m; \theta_k)$ as the accumulated random features for action sequences $\{a_1, \cdots, a_H\}$ taken from state $s_1$. We then use $K$ function approximators representing each of the $K$ Q-basis functions, e.g., neural networks $\psi(\cdot, \cdot; \nu_k) : \mathcal{S} \times \mathcal{A}^H \to \mathbb{R}$ for $k \in [K]$, to fit the accumulated random features. Specifically, we minimize the following loss

$\min_{\{\nu_k\}} \quad \frac{1}{M} \sum_{m \in [M], k \in [K]} \left( \psi(s_1^m, a_{1:H}^m; \nu_k) - \sum_{h \in [H]} \gamma^{h-1} \phi(s_h^m, a_h^m; \theta_k) \right)^2.$

This objective is a Monte Carlo estimate, but we can also leverage dynamic programming to do so in an infinite-horizon way, as we show in Appendix B.1.

### 3.3 Online Planning: Inferring Q-functions with Linear Regression and Planning

Our second phase concerns online planning to transfer to new tasks with arbitrary rewards using our learned Q-basis functions. For a sufficiently expressive set of random non-linear features, any non-linear reward function can be expressed as a linear combination of these features. As we outline below, this combined with linearity of expectation, allows us to express the task-specific Q-function for any particular reward as a linear combination of learned Q-bases of random functions. Therefore, we can obtain the test-time Q-function by solving a *simple* linear regression problem from random features to instantaneous rewards, and use the *same* inferred coefficients to recombine Q-bases to obtain task-specific Q-functions. This inferred Q-function can then be used to plan an optimal sequence of actions. We describe each of these steps below.

#### 3.3.1 Reward Function Fitting with Randomized Features

We first learn how to express the reward function for the new task as a linear combination of the random features. This can be done by solving a linear regression problem to find the coefficient vector $w = [w_1, \cdots, w_K]^\top$ that approximates the new task's reward function as a linear combination of the (non-linear) random features. Specifically, we minimize the following loss

$$w^* = \underset{w}{\text{argmin}} \quad \frac{1}{MH} \sum_{h \in [H], m \in [M]} \left( r(s_h^m, a_h^m) - \sum_{k \in [K]} w_k \phi(s_h^m, a_h^m; \theta_k) \right)^2 + \lambda \|w\|_2^2, \quad (3.1)$$

where $\lambda \geq 0$ is the regularization coefficient, and $r(s_h^m, a_h^m) \sim R_{s_h^m, a_h^m}$. Due to the use of random features, Eq. (3.1) is a *ridge regression* problem, and can be solved efficiently.

Given these weights, it is easy to estimate an approximate multi-step Q-function for the true reward on the new task by linearly combining the Q-basis functions learned in the offline training phase $\{\psi(\cdot, \cdot; \nu_k^*)\}_{k \in [K]}$ according to the *same* coefficient vector $w^*$. This follows from the additive nature of reward and linearity of expectation. In particular, if the reward function $r(s, a) = \sum_{k \in [K]} w_k^* \phi(s, a; \theta_k)$ holds approximately, which will be the case for large enough $K$ and rich enough $\phi$, then the approximate Q-function for the true test-time reward under the sequence $\{a_1, \cdots, a_H\}$ satisfies $Q^H(s_1, a_{1:H}) := \mathbb{E}_{s_{h+1} \sim \mathcal{T}(s_h, a_h)} \left[ \sum_{h \in [H]} \gamma^{h-1} R_{s_h, a_h} \right] \approx \sum_{k \in [K]} w_k^* \psi(s_1, a_{1:H}; \nu_k^*)$, where $\{w_k^*\}_{k \in [K]}$ is the solution to the regression problem (Eq. (3.1)) and $\{\nu_k^*\}_{k \in [K]}$ is the solution to the Q-basis fitting problem described in Section 3.2.

#### 3.3.2 Planning with Model Predictive Control

To obtain the optimal sequence of actions, we can use the inferred approximate Q-function for the true reward $Q^H(s_1, a_{1:H})$ for online planning at each time $t$ in the new task: at state $s_t$, we conduct standard model-predictive control techniques with random shooting [35, 52], i.e., randomly generating $N$ sequences of actions $\{a_1^n, \cdots, a_H^n\}_{n \in [N]}$, and find the action sequence with the maximum Q-value such that $n_t^* \in \text{argmax}_{n \in [N]} \quad \sum_{k \in [K]} w_k^* \psi(s_t, a_{t:t+H-1}^n; \nu_k^*)$.

We note that this finite-horizon planning is an approximation of the actual infinite-horizon planning, which may be enhanced by an additional reward-to-go term in the objective. We provide such an enhancement with details in Appendix B.1, although we found minimal effects on empirical performance. We then execute $a_t^{n_t^*}$ from the sequence $n_t^*$, observe the new state $s_{t+1}$, and replan. We refer readers to Appendix F for a detailed connection of our proposed method to existing work and Appendix A for pseudocode.

## 3.4 Theoretical Justifications

We now provide some theoretical justifications for the methodology we adopt. To avoid unnecessary nomenclature of measures and norms in infinite dimensions, we in this section consider the case that $\mathcal{S}$ and $\mathcal{A}$ are discrete (but can be enormously large). Due to space limitation, we present an abridged version of the results below, and defer the detailed versions and proofs in Appendix D. We first state the following result on the expressiveness of random cumulants.

**Theorem 3.1** (Q-function approximation; Informal). Under standard coverage and sampling assumptions of offline dataset $\mathcal{D}$, and standard assumptions on the boundedness and continuity of random features $\phi(s, a; \theta)$, it follows that with horizon length $H = \widetilde{\Theta}(\frac{\log(R_{\max}/\epsilon)}{1-\gamma})$ and $M = \widetilde{\Theta}(\frac{1}{(1-\gamma)^3 \epsilon^4})$ episodes in dataset $\mathcal{D}$, and with $K = \widetilde{\Omega}((1-\gamma)^{-2}\epsilon^{-2})$ random features, we have that for any given reward function $R$, and any policy $\pi$

$$\|\widehat{Q}_\pi^H(w^*) - Q_\pi\|_\infty \leq \mathcal{O}(\epsilon) + \mathcal{O}\left(\sqrt{\inf_{f \in \mathcal{H}} \mathcal{E}(f)}\right)$$

with high probability, where $\widehat{Q}_\pi^H(s, a; w^*)$ is defined as $\mathbb{E}\left[\sum_{h=1}^H \gamma^{h-1} \sum_{k \in [K]} w_k^* \phi(s_h, a_h; \theta_k) \,|\, s_1 = s, a_1 = a\right]$ for each $(s, a)$ and can be estimated from the offline dataset $\mathcal{D}$; $\inf_{f \in \mathcal{H}} \mathcal{E}(f)$ is the infimum expected risk over the function class $\mathcal{H}$ induced by $\phi$.

Theorem 3.1 is an informal statement of the results in Appendix D.2, which specifies the number of random features, the horizon length per episode, and the number of episodes, in order to approximate $Q_\pi$ accurately by using the data in $\mathcal{D}$, under any given reward function $R$ and policy $\pi$. Note that the number of random features is not excessive and is polynomial in problem parameters. We also note that the results can be improved under stronger assumptions of the sampling distributions $p$ and kernel function classes [41, 29].

Next, we justify the use of multi-step $Q$-functions in planning in a deterministic transition environment, which contains all the environments our empirical results we will evaluate later. Recall that for any given reward $R$, let $Q_\pi$ denote the $Q$-function under policy $\pi$. Note that with a slight abuse of notation, $Q_\pi$ can also be the multi-step $Q$-function (see definition in Section 2), and the meaning should be clear from the input, i.e., whether it is $Q_\pi(s, a)$ or $Q_\pi(s, a_1, \cdots, a_H)$.

**Theorem 3.2.** Let $\Pi$ be some subclass of Markov stationary policies, i.e., for any $\pi \in \Pi$, $\pi : \mathcal{S} \to \Delta(\mathcal{A})$. Suppose the transition dynamics $\mathcal{T}$ is deterministic. For any given reward $R$, denote the $H$-step policy obtained from $H$-step policy improvement over $\Pi$ as $\pi_H' : \mathcal{S} \to \mathcal{A}^H$, defined as $\pi_H'(s) \in \mathrm{argmax}_{(a_1, \cdots, a_H) \in \mathcal{A}^H} \max_{\pi \in \Pi} Q_\pi(s, a_1, \cdots, a_H)$, for all $s \in \mathcal{S}$. Let $V_{\pi_H'}$ denote the value function under the policy $\pi_H'$ (see formal definition in Appendix D.1). Then, we have that for all $s \in \mathcal{S}$

$$V_{\pi_H'}(s) \geq \max_{a_{1:H}} \max_{\pi \in \Pi} Q_\pi(s, a_1, \cdots, a_H) \geq \max_a \max_{\pi \in \Pi} Q_\pi(s, a).$$

The proof of Theorem 3.2 can be found in Appendix D.2. The result can be viewed as a generalization of the generalized policy iteration result in [5] to multi-step action-sequence policies. Taking $\Pi$ to be the set of policies that generate the data, the result shows that the value function of the greedy multi-action policy improves over all the possible $H$-step multi-action policies, with the policy after step $H$ to be any policy in $\Pi$. Moreover, the value function by $\pi_H'$ also improves overall one-step policies if the policy after the first step onwards follows any policy in $\Pi$. This is due to the fact that $\Pi$ (coming from data) might be a much more restricted policy class than any action sequence $a_{1:H}$.

## 4 Experimental Evaluation

In this section, we aim to answer the following research questions: **(1)** Does `RaMP` allow for effective transfer of behaviors across tasks with varying rewards? **(2)** Does `RaMP` scale to domains with high-dimensional observation and action spaces and long horizons? **(3)** Which design decisions in `RaMP` enable better transfer?

### 4.1 Experiment Setup

Across several domains, we evaluate the ability of `RaMP` to leverage the knowledge of shared dynamics from an offline dataset to quickly solve new tasks with arbitrary rewards.

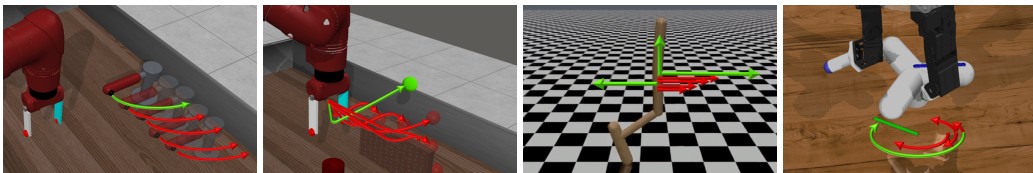

Figure 3: We evaluate our method on manipulation, locomotion, and high dimensional action space environments. The green arrow in each environment indicates the online objective for policy transfer while the red arrows are offline objectives used to label rewards for the privileged dataset.

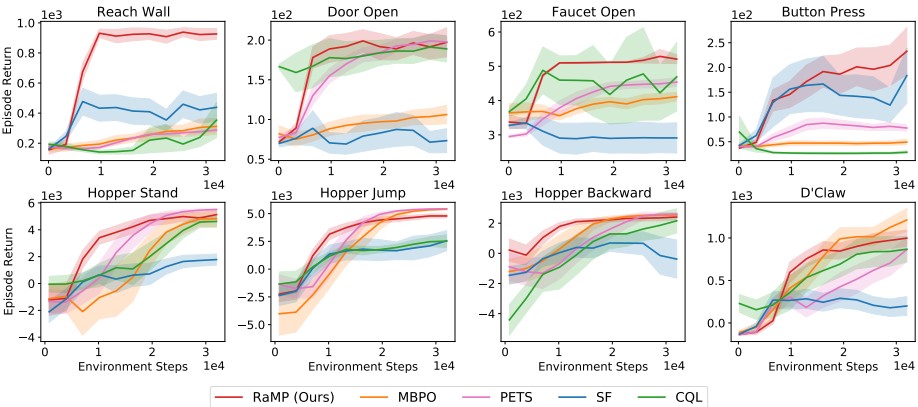

Figure 4: Reward transfer results on Metaworld, Hopper and D'Claw environments. `RaMP` adapts to novel rewards more rapidly than MBPO, Successor Features, and CQL baselines.

**Offline Dataset Construction:** For each domain, we have an offline dataset collected by a behavior policy as described in Appendix C.2. Typically this behavior policy is a mixture of noisy policies accomplishing different objectives in each domain. Although `RaMP` and other model-based methods do not require knowledge of any reward from the offline dataset, other baseline comparisons will require privileged information. Baseline comparison methods like model-free RL and successor features require the provision of a set of training objectives, as well as rewards labeled for these objectives on state actions from the offline dataset. We call such objectives "offline objectives" and a dataset annotated with these offline objectives and rewards a privileged dataset.

**Test-time adaptation:** At test-time, we select a novel reward function for online adaptation, referred to as 'online objective' below. The online objective may correspond to rewards conditioned on different goals or even arbitrary rewards, depending on the domain. Importantly, the online objective need not be drawn from the same distribution as the privileged offline objectives.

Given this problem setup, we compare `RaMP` with a variety of baselines. (1) MBPO [19] is a model-based RL method that learns a standard one-step dynamics model and uses actor-critic methods to plan in the model. We pre-train the dynamics model for MBPO on the offline dataset. (2) PETS [8] is a model-based RL method that learns an ensemble of one-step dynamics models and performs MPC. We pre-train the dynamics model on the offline dataset and use the cross-entropy method (CEM) to plan. (3) Successor feature (SF) [5] is a framework for transfer learning in RL as described in Sec. 1.1. SF typically assumes access to a set of policies towards different goals along with a learned featurization, so we provide it with the privileged dataset to learn a set of training policies. We also learn successor features with the privileged dataset. (4) CQL [24]: As an *oracle* comparison, we compare with a goal-conditioned variant of an offline RL algorithm (CQL). CQL is a model-free offline RL algorithm that learns policies from offline data. While model-free offline RL naturally struggles to adapt to arbitrarily changing rewards, we provide CQL with information about the goal at both training and testing time. CQL is trained on the distribution of training goals on the offline dataset, and finetuned on the new goals at test time. The CQL comparison is assuming access to more information than `RaMP`. Each method is benchmarked on each domain with 4 seeds.

### 4.2 Transfer to Unseen Rewards

We first evaluate the ability of `RaMP` to learn from an offline dataset and quickly adapt to unseen test rewards in 4 robotic manipulation environments from Meta-World [56]. We consider skills like

reaching a target across the wall, opening a door, turning on a faucet, and pressing a button while avoiding obstacles, which are challenging for typical model-based RL algorithms (Fig. 3). Each domain features 50 different possible goal configurations, each associated with a different reward but the same dynamics. The privileged offline objectives consist of 25 goal configurations. The test-time reward functions are drawn from the remaining 25 "out-of-distribution" reward functions. We refer the reader to Appendix C.1 for details.

As shown in Fig 4, our method adapts to test reward most quickly across all four meta-world domains. MBPO slowly catches up with our performance with more samples, since it still needs to learn the Q function from scratch even with the dynamics branch trained. PETS adapts faster than MBPO but performs worse than our method as it suffers from compounding error. In multiple environments, successor features barely transfer to the online objective as it entangles policies that are not close to those needed for the online objective. Goal-conditioned CQL performs poorly in all tasks as it has a hard time generalizing to out-of-distribution goals. In comparison, RaMP is able to deal with arbitrary sets of test time rewards, since it does not depend on the reward distribution at training time.

### 4.3  Scaling to Tasks with Longer Horizons and High Dimensions

We further evaluate the ability of our method to scale to tasks with longer horizons. We consider locomotion domains such as the Hopper environment from OpenAI Gym [7]. We chose the offline objectives to be a mixture of skills such as standing, sprinting, jumping, or running backward with goals defined as target velocities or heights. The online objectives consist of unseen goal velocities or heights for standing, sprinting, jumping or running. As shown in Fig. 4, our method maintains the highest performance when adapting to novel online objectives, as it avoids compounding errors by directly modeling accumulated random features. MBPO adapts at a slower rate since higher dimensional observation and longer horizons increase the compounding error of model-based methods. We note that SF is performing reasonably well, likely because the method also reduces the compounding error compared to MBPO, and it has privileged information.

To understand whether RaMP can scale to higher dimensional state-action spaces, we consider a dexterous manipulation domain (referred to as the D'Claw domain in Fig 3). This domain has a 9 DoF action space controlling each of the joints of the hand as well as a 16-dimensional state space including object position. The offline dataset is collected moving the object to different orientations, and the test-time rewards are tasked with moving the object to new orientations (as described in Appendix C.1). Fig 4 shows that both Q-estimation and planning with model-predictive control remain effective when action space is large. In Appendix C.3, we test our method on environments with even higher dimensional observations such as images.

### 4.4  Ablation of Design Choices

To understand what design decisions in RaMP enable better transfer and scaling, we conduct ablation studies on various domains, including an analytical 2D point goal-reaching environment and its variants (described in Appendix C.1), as well as the classic double pendulum environment and meta-world reaching. We report an extensive set of ablations in Appendix C.5.

**Reduction of compounding error with multi-step Q functions**

We hypothesize that our method does not suffer from compounding errors in the same way that feedforward dynamics models do.

In Table 1, we compare the approximation error of truncated Q values computed with (1) multi-step Q functions obtained as a linear combination of random cumulants (Ours), and (2) rollouts of a dynamics model (MBRL). We train the methods on the offline data and evaluate on the data from a novel task at test time. As shown in Table 1, our method outperforms feedforward dynamics models.

|      | Hopper | Pendulum |
|------|--------|----------|
| Ours | **25.7 $\pm$ 5.4** | **65.1 $\pm$ 0.4** |
| MBRL | 50.2 $\pm$ 9.7 | 395.4 $\pm$ 5.8 |

Table 1: Policy evaluation error. Feedforward dynamics model suffers from compounding error that is particularly noticeable in domains with high action dimensions or chaotic dynamics. Our method achieves low approximation error in both domains.

## 5  Discussion

In this work, we introduce RaMP, a method that leverages diverse unlabeled offline data to learn models of long horizon dynamics behavior, while being able to naturally transfer across tasks with different reward functions. To this end, we propose to learn the long-term evolution of random features

under different action sequences, which can be generated in a self-supervised fashion without reward labels. This way, we are able to disentangle dynamics, rewards, and policies, while without explicitly learning the dynamic model. We validate our approach across a number of simulated robotics and control domains, with superior transfer ability than baseline comparisons. There are also limitations of the current approach we would like to address in the future work: the current Monte-Carlo-based value function estimation may suffer from high variance, and we hope to incorporate the actor-critic algorithms [23] in `RaMP` to mitigate it; current approach is only evaluated on simulated environments, and we hope to further test it on real robotic platforms.

## Acknowledgments

We would like to thank Russ Tedrake, Max Simchowitz, Anurag Ajay, Marius Memmel, Terry Suh, Xavier Puig, Sergey Levine, Seohong Park and many other members of the Improbable AI lab at MIT and the WEIRD Lab at University of Washington for valuable feedback and discussions.

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

# Supplementary Materials for "Self-Supervised Reinforcement Learning that Transfers using Random Features"

## A    Algorithm Pseudocode

---

**Algorithm 1** Model-Free Transfer with Randomized Cumulants and Model-Predictive Control

---

1: **Input**: Offline dataset $\mathcal{D}$ given in Section 2, distribution $p$ over $\mathbb{R}^d$, number of random features $K$

2: **Offline Training Phase:**

3: Randomly sample $\{\theta_k\}_{k \in [K]}$ with $\theta_k \sim p$, and construct dataset

$$\mathcal{D}_\phi = \left\{ \left( (s_1^m, a_{1:H}^m), \sum_{h \in [H]} \gamma^{h-1} \phi(s_h^m, a_h^m; \theta_k) \right) \right\}_{m \in [M], k \in [K]}.$$

4: Fit random Q-basis functions $\psi(\cdot, \cdot, \nu_k) : \mathcal{S} \times \mathcal{A}^H \to \mathbb{R}$ for $k \in [K]$ by minimizing the loss over the dataset $\mathcal{D}_\phi$,

$$\left\{ \nu_k^* \right\}_{k \in [K]} \in \underset{\{\nu_k\}_{k \in [K]}}{\operatorname{argmin}} \ \frac{1}{M} \sum_{m \in [M], k \in [K]} \left( \psi(s_1^m, a_{1:H}^m; \nu_k) - \sum_{h \in [H]} \gamma^{h-1} \phi(s_h^m, a_h^m; \theta_k) \right)^2.$$

5: **Online Planning Phase:**

6: Fit the testing task's reward function $r(\cdot, \cdot)$ with linear regression on random features:

$$w^* \in \underset{w}{\operatorname{argmin}} \ \frac{1}{MH} \sum_{h \in [H], m \in [M]} \left( r(s_h^m, a_h^m) - \sum_{k \in [K]} w_k \phi(s_h^m, a_h^m; \theta_k) \right)^2 + \lambda \|w\|_2^2$$

   where $r(s_h^m, a_h^m) \sim R_{s_h^m, a_h^m}$.

7: Sample $s_1 \sim \mu_0$

8: **for** time index $t = 1, \cdots$ **do**

9:     Randomly generate $N$ sequences of actions $\{a_1^n, \cdots, a_H^n\}_{n \in [N]}$

10:    Find the best sequence such that

$$n_t^* \in \underset{n \in [N]}{\operatorname{argmax}} \ \sum_{k \in [K]} w_k^* \psi(s_t, a_{t:t+H-1}^n; \nu_k^*).$$

    Execute $a_t^{n_t^*}$ from the sequence $n_t^*$, observe the new state $s_{t+1} \sim \mathcal{T}(s_t, a_t^{n_t^*})$

11: **end for**

---

## B    Implementation Details

Our implementation contains a few optional components that are omitted from main paper due to page limit. We note that RaMP's performance already surpasses the baselines even without them. These components are introduced for the completeness of our main algorithm and leads to better sample efficiency in different situations.

### B.1    Infinite-Horizon $Q$-function Variant

While our setup in Section 2 uses a finite-horizon $Q_\pi^H$ to approximate $Q_\pi$, our method can also plan with an infinite-horizon $Q$ function during the online phase. In this section, we describe one compatible way to learn an infinite-horizon $Q$-function while still enjoying the benefits of RaMP in the case with *deterministic* transition dynamics. We find this variant leads to continual improvement and thus better performance in a few longer-horizon environments.

We first notice that an infinite-horizon $Q$-function $Q_\pi$ can be decomposed into the discounted sum of an $H$-step reward and a discounted value function under policy $\pi$ evaluated at $s_{t+H}$:

$$Q_\pi(s', a') = \mathbb{E}_{\substack{a_t \sim \pi(\cdot \mid s_t) \\ s_{t+1} \sim \mathcal{T}(\cdot \mid s_t, a_t)}} \left[ \gamma^H V_\pi(s_{H+1}) + \sum_{t=1}^{H} \gamma^{t-1} r(s_t, a_t) \,\Big|\, s_1 = s', a_1 = a' \right]$$

$$\text{where } V_\pi(s') = \mathbb{E}_{a_t \sim \pi(\cdot \mid s_t)} \left[ \sum_{t=1}^{\infty} \gamma^{t-1} r(s_t, a_t) \,\Big|\, s_1 = s' \right].$$

Given a policy $\pi$, value function $V_\pi^\theta$ parameterized by $\theta$ can be learned via gradient descent and Monte-Carlo method:

$$\theta \leftarrow \theta - \alpha \nabla_\theta \, ||V_\pi^\theta(s_t) - (r_t + \gamma V_\pi^{\theta'}(s_{t+1}))||_2^2 \text{, for sampled } (s_{t:t+1}, a_t, r_t) \sim \tau_\pi$$

where $\tau_\pi$ is trajectory rollouts collected with current policy $\pi$ and $\theta'$ is a target network that gets updated by $\theta$ with momentum.

Now consider our multi-step setup. Our multi-step $Q$-function can also be written as the sum of our $H$ step approximation and discounted value function at $s_{H+1}$:

$$Q_\pi(s, \widetilde{a}_{1:H}) = \mathbb{E}_{\substack{a_{H+t} \sim \pi(\cdot \mid s_{H+t}) \\ s_{t+1} \sim \mathcal{T}(\cdot \mid s_t, a_t)}} \left[ \sum_{t=1}^{\infty} \gamma^{t-1} r_t \,\Big|\, s_1 = s, a_1 = \widetilde{a}_1, \cdots, a_H = \widetilde{a}_H \right]$$

$$= Q_\pi^H(s, \widetilde{a}_{1:H}) + \gamma^H V_\pi(s_{H+1}),$$

where we note that in the last line, there is no expectation over $s_{H+1}$ since the transition dynamics is deterministic, and $s_{H+1}$ is deterministically decided by $(s_1, a_{1:H})$.

Vanilla RaMP enables efficient estimation of $Q_\pi$ with novel reward function at the cost of truncating the second term above with $Q_\pi \approx Q_\pi^H$. As we have shown in Section 4, planning with this finite-horizon $Q$-approximation would already lead to reasonable planning in most of the experiments.

We can go a step further and also estimate the second term so we can plan in an infinite-horizon setting. The main challenge is getting $V_\pi(s_{H+1})$ in our multi-step setup, as we don't explicitly predict $s_{H+1}$. This, however, can be addressed easily by reparameterizing $V_\pi(s_{H+1})$ on an action sequence that leads to $s_{H+1}$ just like what we did for $Q$. We thus define a multi-step value function

$$F_\pi(s, \widetilde{a}_{1:H}) = \mathbb{E}_{s_{H+1}} \left[ V_\pi(s_{H+1}) \,\big|\, s_1 = s, a_1 = \widetilde{a}_1, \cdots, a_H = \widetilde{a}_H \right].$$

Then $Q_\pi(s, a_{1:H}) = Q_\pi^H(s, a_{1:H}) + \gamma^H \cdot F_\pi(s, a_{1:H})$. Under our deterministic transition dynamics, $s_{H+1}$ is fully determined by $(s_1, \widetilde{a}_{1:H})$, so we can remove the expectation in the equation. We then rewrite the training objective $V_\pi$ in terms of $F_\pi$ to learn $F_\pi$:

$$\theta \leftarrow \theta - \alpha \nabla_\theta \, ||V_\pi^\theta(s_{t+H}) - (r_{t+H} + \gamma V_\pi^{\theta'}(s_{t+H+1}))||_2^2$$
$$\text{for sampled } (s_{t+H:t+H+1}, a_{t+H}, r_{t+H}) \sim \tau_\pi$$

becomes

$$\theta \leftarrow \theta - \alpha \nabla_\theta \, ||F_\pi^\theta(s_t, a_{t:t+H-1}) - (r_{t+H} + \gamma F_\pi^{\theta'}(s_{t+1}, a_{t+1:t+H}))||_2^2$$
$$\text{for sampled } (s_{t:t+1}, a_{t:t+H}, r_{t+H}) \sim \tau_\pi$$

where we parameterize multi-step value function $F_\pi$ by $F_\pi^\theta$. So we can learn $F_\pi$ with Monte-Carlo sampling and gradient descent just like what we do to learn $V_\pi$ in the single-step case above. Combined with $Q_\pi^H$, we now have an estimation for the infinite-horizon $Q$-function $Q_\pi$ in a multi-step manner.

For planning, we do on policy learning by alternating between policy rollout and $Q_\pi$ learning. As a policy, MPC planner first uses the infinite-horizon $Q_\pi(s, a_{1:H}) = Q_\pi^H(s, a_{1:H}) + \gamma^H \cdot F_\pi(s, a_{1:H})$ to plan and collect rollouts. Then $F_\pi$ is trained with these on policy rollouts while $Q_\pi^H$ is also learned like vanilla RaMP via online least square. By incorporating this infinite-horizon variant, our MPC planner can now plan with an infinite horizon during the online phase.

Infinite-horizon Q-function variant is used in all benchmarks in Figure 4.

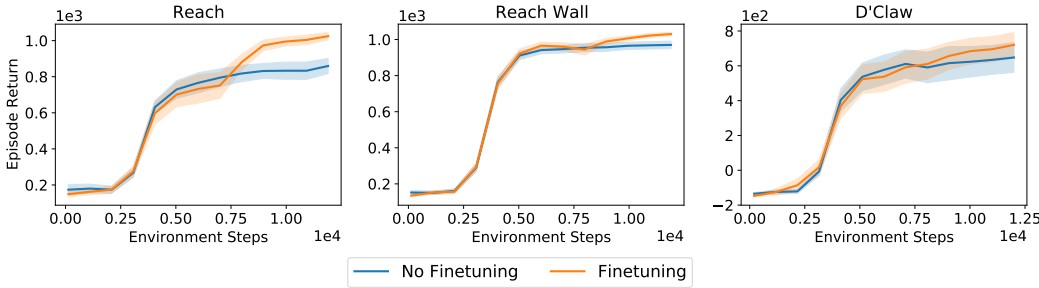

Figure 5: Finetuning results on MetaWorld and D'Claw. Our method is able to continuously improve by finetuning the Q-basis networks during online training.

## B.2    Online Psi Finetuning

While we freeze the Q-basis networks during online training in our setup in Section 2, we can also finetune the Q-basis networks to continuously improve our estimate of the Q-value. After performing reward regression for a number of steps, we can finetune the Q-basis networks on online trajectories by fitting the predicted Q-values to the Monte-Carlo Q values and allowing the gradients to flow through the regression weights. To illustrate the effectiveness of finetuning, we train our algorithm on a set of environments for 12800 steps, where we freeze the regression weights and begin finetuning after 6400 environment steps. As shown in Fig. 5, our method sees a noticeable performance increase with finetuning. We incorporate finetuning in our Hopper and D'Claw experiments, where we finetune the psi network every 800 steps.

## B.3    Planning With Uncertainty

Since the Q-basis networks are trained on an offline dataset, they can provide inaccurate Q estimates when online planning goes out of distribution. To encourage the agent to stay within distribution, we train an ensemble of Q-basis networks and augment the online planning objective with a disagreement penalty. Let $\{\psi_m\}_{m\in[M]}$ denote the ensemble of Q-basis networks. The planning objective becomes

$$n_t^* \in \operatorname*{argmax}_{n\in[N]} \frac{1}{M} \sum_{m\in[M]} \sum_{k\in[K]} w_k^* \psi_m(s_t, a_{t:t+H-1}^n; v_k^*) - \beta \operatorname{Var}\left(\left\{\sum_{k\in[K]} w_k^* \psi_m(s_t, a_{t:t+H-1}^n; v_k^*)\right\}_{m\in[M]}\right)$$

(B.1)

where $\beta$ is a hyperparameter controlling the weight of the penalty. We use $\beta = 1$ for all our experiments.

## B.4    Model-Predictive Path Integral (MPPI)

Our method benefits from more powerful planning algorithms such as model-predictive path integral (MPPI) [53]. MPPI is an sampling-based trajectory optimization method which maintains a distribution of action sequence initialized as isotropic standard Gaussian. In each iteration, MPPI draws $n$ action sequences from the current distribution and computes their value using the learned Q-basis networks and online regression weights. It then updates the action distribution using a weighted mean and standard deviation of the sampled trajectories, where the weights are computed as the softmax of the values multiplied by a temperature parameter $\gamma$. We use an MPPI planner with $\gamma = 10$ for the D'Claw environment.

## C    Setup Details and Additional Experiments

In this section, we provide more details of the experiments, including more detailed setup and supplementary results.

## C.1 Description of Environments

We describe the details of all used environments such as observation space, action space, reward, offline / online objectives, and dataset collection.

**Meta-World** All our Meta-World [56] domains share the standard Meta-World observation which includes gripper location, and object locations of all possible objects involved in the Meta-World benchmark. Although the observation space has 39 dimensions, each domain only uses one or two objects so only 7 dimensions are changing any each domain we chose. For pixel observation variants of each domain, we concatenate two $84 \times 84 \times 3$ RGB images from two views, with a resulting observation dimension of 42336. Each domain has a 4 dimensional action space, corresponding to the delta movement of end-effector in the workspace along with the delta movement of the gripper finger. Metaworld provides a set of 50 goal configurations for each domain. We collect offline dataset following the procedure described in Sec. C.2. The online objective is chosen to be a novel configuration that isn't any of the 50 offline goal configurations. To create the privileged dataset, we choose 25 of the goal configurations as offline objectives. These chosen configurations are the furthest 25 goals from the online objective in the Euclidean distance. We evenly annotate the offline dataset with rewards for each of these goals to form a privileged dataset such that the online objective is out of the distribution of the offline objectives. For different configurations of the same domain, since object locations are in observation and the goal configuration isn't in it, the dynamics is the same. We now describe the objectives of all used meta-world domains, including those used in the appendix.

1. **Reach across Wall** The objective is to reach a target across a wall. The location of the target is randomized for each goal configuration.

2. **Open Door** The objective is to open the door to a certain angle by putting the end-effector between the door handle and the door surface. For each configuration, the location of the door is randomized.

3. **Turn on Faucet** The objective is to turn on a faucet by rotating it along an axis. For each goal configuration, the location of the faucet is randomized.

4. **Press Button** The objective is to press a button horizontally. The location of the button is randomized for each goal configuration.

**Hopper** Hopper is an environment with a higher dimensional observation of 11 dimensions and an action space dimension of 3. Hopper is a locomotion environment that requires long-horizon reasoning since a wrong action will make it fall down and touch the ground only after some steps. The objective of the original Hopper enviroment in OpenAI gym is to train it to run forward. To analyze the performance of CQL and SF on Hopper, we modify the objective to a goal-conditioned variant such that the agent is trained to follow a certain velocity specified by its goal. Similar modifications are common in meta reinforcement learning such as in [12]. We sample a set of 50 goal velocities between 0.5 and 0.75 as offline objectives to collect data in the same way as we did in Metaworld environment. To ensure good coverage of state-action space, we collect data following a set of 50 $\epsilon$-noised expert policies. For online transfer, we choose four domains with four different online objectives. All the variant domains of Hopper share the same dynamics.

1. **Hopper Stand** The objective is to stand still and upright. This online objective is on the boundary of the offline objectives' range.

2. **Hopper Jump** The objective is to jump to a height of 1.5, without moving forward or backward. Such height is typically not achievable when running forward, so this objective is drastically different from offline objectives.

3. **Hopper Backward** The objective is to run backward at a target speed of 0.5, which is in the opposite direction of the offline objective. Falling down to the ground is penalized.

4. **Hopper Sprint** The objective is to run forward at a speed of 1.5, which is out of the distribution of offline objectives ranging from 0 to 1.0. The reward function remains the same as that for offline objectives. Only the goal changes.

The 50 $\epsilon$-noised expert policies used to collect data contains 10 policies from Hopper Stand, 10 policies from Hopper Jump, 10 policies from Hopper Backward, 10 policies from Hopper Sprint and

10 policies from a another hopper environment whose goal is running at a random velocity between 0 and 1. We note the 50 policies doesn't provide any information about their corresponding rewards because the reward is only computed towards the different set of goal between 0.5 and 0.75.

**D'Claw**  D'Claw environment is a dexterous manipulation environment with 24 dimensions of observation space and 9 dimensions of action space. The 9 dimensions correspond to the 9 joints of a robot hand with 3 fingers and 3 joints on each finger. The objective of the environment is to control the hand to rotate a tripod to a certain angle. Angular distance to the target angle is used as the offline objective. To increase the degree of freedom and make the environment more challenging, we allow the tripod to not only rotate but also translate freely on the 2d plane. The initial rotation and position of thetripod are randomized upon every episode. We collect the offline dataset in the same way as in meta-world, training on 50 offline objectives and using $\epsilon$-greedy to collect rollouts. At test time we choose a new offline objective angle and annotate the rewards of the privileged dataset in the same way we did for goal conditioned Metaworld environments.

**Point**  Point is an analytical 2D goal-reaching environment with linear dynamics. The reward is defined as the distance to goal minus an action penalty. The offline objectives are negative distances to the randomly selected goals on the 2D plane. The online objectives are novel goals on the plane. Since we are not evaluating CQL and SF on this environment, we don't generate the privileged dataset for Point.

**Point Perturbed**  Point Perturbed shares the same linear dynamics as Point, but features unsafe regions with negative rewards or local maxima with small positive rewards. These perturbations represent out-of-distribution test objectives that cannot be well approximated by a low-dimensional feature vector. Note that the added perturbations make Point Perturbed no longer an instance of the Point environment with a different goal. Instead, it features online objectives that are completely in a different class from Point.

## C.2  Training Details

For each domain, we first train 50 policies with SAC [15]. Each policy is trained towards some offline objective of the domain described in Sec. C.1 for 50000 steps. We then use an $\epsilon$-greedy version of each trained policy to collect 32000 data points for each domain per offline objective. We choose $\epsilon = 0.5$. Such a procedure ensures the dataset has reasonable coverage of the entire state-action space. We note training these policies are fully optional, since RaMP only trajectories without rewards. Datasets collected via intrinsic rewards like curiosity would totally suffice. We also note that the environment used to train expert policies are not necessarily the same as environments used to collect the data points. They should share the same dynamics yet the only the later provides reward information to our offline dataset. In particular, all metaworld data are collected via their original goal conditioned policies while hopper data are collected with a mixture of running and jumping policies in a different goal-velocity-conditioned environment as decribed in Sec. C.1. We choose the random feature dimension to be 2048. Each dimension in the random feature is extracted by feeding the state-action tuple to a randomly initialized MLP with 2 hidden layers of size of 32. There are therefore 2048 independent, small random MLPs to extract $\phi$. All state-action tuples are projected to reward basis $\phi$ with it.

During offline training phase, we ensemble 8 instances of MLP with 2 hidden layers of size 4096 and train $\psi$ network following Sec. 3.2. We train $\psi$ network with a learning rate of $3 \times 10^{-4}$ on the offline dataset for 4 epochs, with a $\gamma$ decay of 0.9 and batch size 128. We choose the horizon $H$ to be 16 for Meta-World and Hopper environments and 10 for D'Claw. During online adaptation phase, we first do random exploration for 2500 steps to collect enough data points for linear regression. When doing linear regression, we always concatenate a bias dimension to $\psi$. Since online least square makes recomputing $\omega$ regression fast, we perform an update of weight vector every single step after initial random exploration is finished. In each MPC step, we randomly sample 1024 action sequences. We penalize the predicted rewards with the variance of predictions from all 8 ensembles following Sec. B.3.

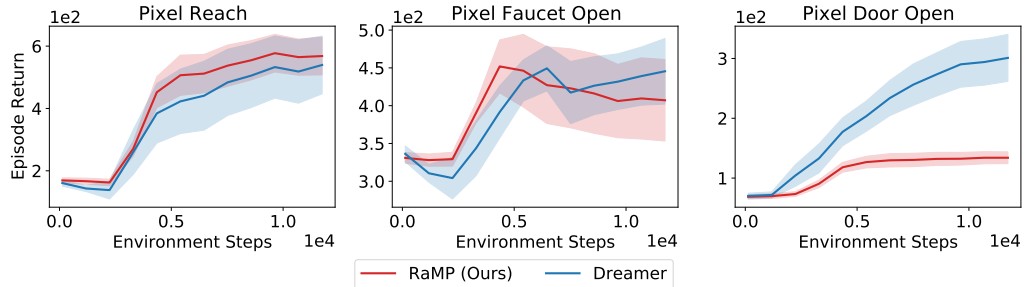

Figure 6: Results on Metaworld with high-dimensional pixel observation. `RaMP` achieves comparable performance to Dreamer on Pixel Reach and Pixel Faucet Open but struggles to perform well on Pixel Door Open. The performance of `RaMP` on Pixel Faucet Open and Pixel Reach is very similar to its performance in state observation environments.

## C.3  Scaling to High-dimensional Pixel Observation

In Sec.4.3, we evaluate `RaMP`'s ability to scale to environments with high dimensional observations. In this section, we go a step further by evaluating `RaMP` on visual domains, effectively increasing the dimension of the observation space to 42336 as described in Sec. C.1. We use a CNN encoder following the architecture in [32] followed by 2 layer MLP as the random feature network. Both CNN and MLP layers are randomly initialized. Action is projected and concatenated to the first layer of MLP so the random feature would still be conditioned on both action and observation. We compare our method against the Dreamer [16], the state of art model-based learning method optimized for pixel observations. Similar to MBPO, we pre-train dreamer's dynamics branch with offline data before the online phase. As shown in Fig. 6, our method is able to achieve a similar level of performance as Dreamer in two meta-world environments, Pixel Reach and Pixel Faucet Open. `RaMP` does not see a significant return drop from the variant of the environment with state observation. Given the efficacy of Dreamer, the result still shows `RaMP`'s performance can scale up to pixel observation. However, Dreamer is able to outperform `RaMP` significantly in an environment like Pixel Door Open. This is likely because random features capture the change in input space rather than reward space. Pixel observations can give random features a lot of noise while important semantic features may not correspond to a big change in pixel space. We note that instead of using random convolution layers, we can use pre-trained encoders to achieve better semantic feature extraction and significantly improve the quality of random features. This is beyond the scope of our work and we leave this for future works.

## C.4  Additional Results on Hopper Sprint

Due to page limit, we omitted the plot for Hopper Sprint in Fig.4. Here we provide additional results of `RaMP` and baselines for it in Fig. 7. The result is consistent with our analysis in the main paper. `RaMP` matches or outperforms the baselines just as in other Hopper variants. One major difference here is that CQL is performing well for HopperSprint. This is likely because the online objective of Hopper Sprint is at the boundary of the offline objectives as described in Sec. C.1. Given that offline objectives are running at target velocities, the CQL likely learns to not fall down even if the online objective is out of distribution. By not falling down alone, CQL is capable of maintaining a good reward as seen in this case.

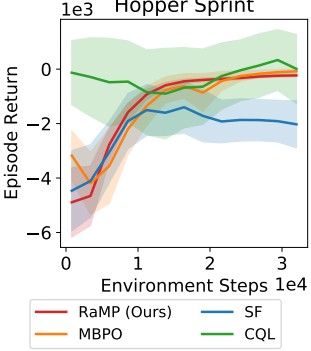

Figure 7: Results on Hopper Sprint

## C.5  Additional Ablations

Our method builds on the assumption that a linear combination of high dimensional random features can approximate arbitrary reward functions. In the following ablation experiments, we validate this assumption both quantitatively and qualitatively.

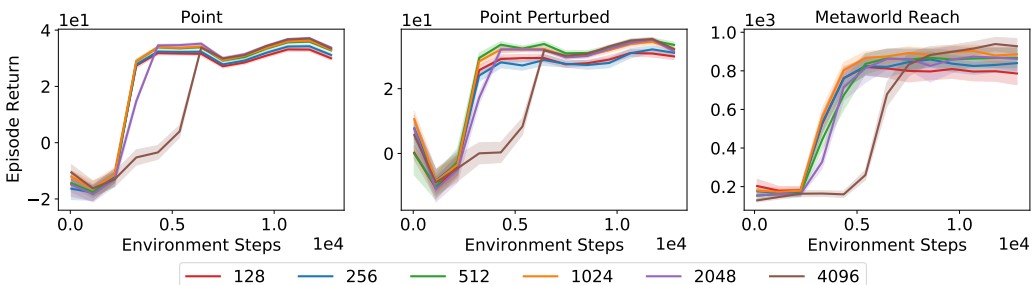

Figure 8: Learning curves for different random feature dimensions. Low-dimensional random features suffer from poor convergence performance, whereas high-dimensional random features experience slow adaptation.

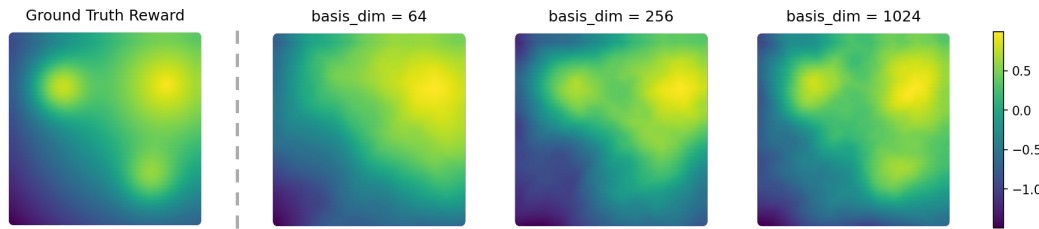

Figure 9: Visualization of reward approximated with different numbers of random basis. As the number of random basis increases, the reward approximation becomes more accurate.

**Effect of different types of featurization** We experiment with three choices of projections: random features parameterized by a deep neural network, random features parameterized by a gaussian matrix, and polynomial features of state and action up to the second order. We evaluate these choices on Point and MetaWorld Reach in Table 2. We see that a linear combination of NN-parametrized random features approximate the true reward well in all tasks. Polynomial features perform well on environments with simple rewards, but struggle as the reward becomes more complex, while Gaussian features are rarely expressive enough.

**Effect of random feature dimension** We evaluate our method on Point, Point Perturbed, and Metaworld Reach using $\{128, 256, 512, 1024, 2048, 4096\}$ random features. We summarize the final returns in Table. 3 and plot the learning curves in Fig. 8. We find that performance degrades as the feature dimension gets lower because the random features are less capable of approximating the true reward. On the other hand, higher-dimensional random features experience slower adaptation. In addition, we provide visualizations of reward approximation in Fig. 9.

|  | Point | Reach |
|---|---|---|
| Random | **34.0 ± 1.0** | **820.8 ± 142.0** |
| Gaussian | -3.6 ± 7.4 | 188.6 ± 49.2 |
| Polynomial | 24.5 ± 5.8 | 162.9 ± 36.4 |

Table 2: Return for different features. Random features are able to approximate the true reward well across domains. Polynomial features work in simple environments but do not scale to complex rewards. Gaussian features are unable to express the reward.

**Scaling with state dimension** In Table 4 we evaluate our method on analytical point environments with 2, 3, and 4 state dimensions. All three environments feature distance-to-goal with an action penalty as the reward. We compute the return error stemming from linear regression as well as the Q error stemming from both linear regression and function approximation. We see an increase in both return error and Q error with higher state dimensions, but overall the approximation errors remain reasonably low.

**Nonlinear approximation capability** In Fig. 10, we visualize the truncated Q value obtained from a linear combination of the random cumulants and compare it to the ground truth Q value approximated by Monte-Carlo sampling. We perform the comparison in Point and Point Perturbed environments.

Table 3: Return as a function of random feature dimension. Low dimensional random features are unable to approximate the true reward with linear regression, leading to degraded convergence performance.

|  | Point | Point Perturbed | Metaworld Reach |
|---|---|---|---|
| 128 | $30.46 \pm 1.62$ | $29.61 \pm 2.55$ | $765.38 \pm 129.98$ |
| 256 | $31.53 \pm 1.45$ | $31.09 \pm 0.04$ | $806.12 \pm 104.77$ |
| 512 | $33.02 \pm 1.10$ | $\mathbf{33.13 \pm 2.09}$ | $824.56 \pm 121.52$ |
| 1024 | $33.36 \pm 1.10$ | $32.02 \pm 2.38$ | $\mathbf{855.14 \pm 85.23}$ |
| 2048 | $\mathbf{34.01 \pm 1.04}$ | $31.83 \pm 1.31$ | $830.84 \pm 142.02$ |
| 4096 | $33.31 \pm 1.15$ | $32.05 \pm 1.11$ | $802.45 \pm 94.47$ |

Table 4: Approximation error with different state dimensions. As state dimension increases, approximation error increases but remains in a reasonable range.

|  | Point 2D | Point 3D | Point 4D |
|---|---|---|---|
| Return Error | $0.42 \pm 0.20$ | $1.20 \pm 0.61$ | $1.32 \pm 0.70$ |
| Q Error | $0.32 \pm 0.05$ | $0.51 \pm 0.10$ | $0.61 \pm 0.12$ |

Our method provides an accurate estimate of the Q value even in the face of out-of-distribution and highly nonlinear rewards.

**In-distribution baselines** Finally, we compare the performance of RaMP to baselines on in-distribution online objectives. Our method is designed to make no assumptions about test objectives during policy transfer. As shown in Fig. 4, RaMP outperforms CQL and SF when the online objectives are out of distribution. A natural question to ask is how things will change when the setting satisfies the assumptions of offline-online objective correlation. For example, in a 2D reaching environment, the training dataset may be annotated with either rewards corresponding to only goals on the right half or goals covering the entire plane. When the online objective is to reach a goal on the left half of the plane, it will be out of distribution for the first case while being in distribution for the second. As shown in Fig. 11, when we curate the labeling process of the privileged dataset to satisfy the in-distribution assumption, CQL and SF receive a significant performance boost, while the performance

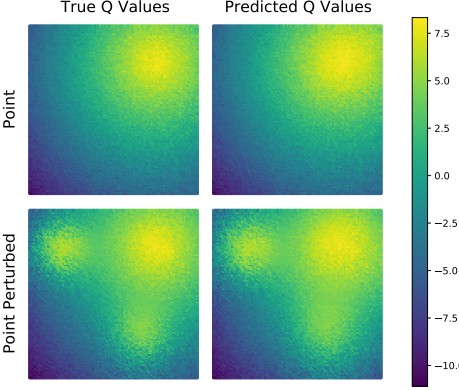

Figure 10: Visualization of true Q value and approximated Q value. Our method is able to approximate the Q value in the face of out-of-distribution and highly nonlinear rewards.

of our method and MBPO are unaffected as neither algorithm depends on the offline objectives. This serves as a foil to the generalization of our method to out-of-distribution online objectives.

## D    Detailed Theoretical Results

In this section, we provide the formal statement of the theoretical insights given in §3.4, and corresponding proofs.

### D.1    Formal Statement

**Theorem D.1.** Suppose the offline data in $\mathcal{D}$ are generated from some distribution $\rho \in \Delta(\mathcal{S} \times \mathcal{A})$, i.e., $(s_h^m, a_h^m) \sim \rho(\cdot, \cdot)$ and $s_{h+1}^m \sim \mathcal{T}$ for all $(m, h) \in [M] \times [H]$, and $\inf_{(s,a) \in \mathcal{S} \times \mathcal{A}} \rho(s, a) = \underline{\rho} > 0$. Suppose $\theta_k \sim p(\cdot)$ for all $k \in [K]$, and $\sup_{(s,a,\theta) \in \mathcal{S} \times \mathcal{A} \times \mathbb{R}^d} |\phi(s, a; \theta)| \leq \kappa$ for some $\kappa > 0$ and

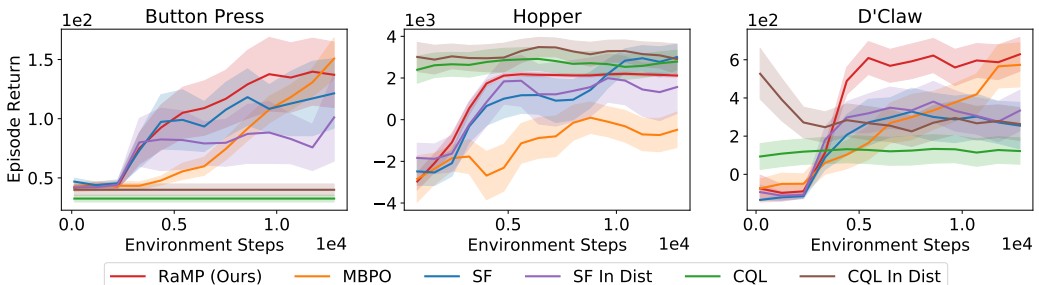

Figure 11: In-distribution results for Successor Features and CQL. Note that `RaMP` and MBPO are unaffected since they do not depend on the distribution of the offline objectives.

$\phi(\cdot, \cdot; \theta)$ is continuous. For some large enough $n := MH$, letting $\lambda = n^{-1/2}$, we have that if $K = \Omega(\sqrt{n}\log(\kappa^2\sqrt{n}/\delta))$, then with probability at least $1 - \delta$, for any given reward function $R$

$$\|\widehat{Q}_\pi(w^*) - Q_\pi\|_\infty \leq \frac{1}{1-\gamma}\sqrt{\frac{1}{\rho}\left[\inf_{f\in\mathcal{H}}\underbrace{\sum_{(s,a)\in\mathcal{S}\times\mathcal{A}}\int\left(r - f(s,a)\right)^2 dR_{s,a}(r)\rho(s,a)}_{\mathcal{E}(f)} + \mathcal{O}\left(\frac{\log(1/\delta)}{\sqrt{n}}\right)\right]}$$

for any policy $\pi$, where $\mathcal{H} := \{f = \int \phi(\cdot, \cdot; \theta)w(\theta)dp(\theta) \mid \int |w(\theta)|^2 dp(\theta) < \infty\}$, $w^*$ is the solution to (3.1), and

$$\widehat{Q}_\pi(s, a; w^*) := \mathbb{E}\left[\sum_{h=1}^\infty \gamma^{h-1}\sum_{k\in[K]} w_k^*\phi(s_h, a_h; \theta_k) \,\Big|\, s_1 = s, a_1 = a\right]. \tag{D.1}$$

The proof of Theorem D.1 can be found in §D.2. It shows that with large enough amount of random features, the $Q$-function of any reward function $R$, under any policy $\pi$, can be approximated accurately up to some inherent error related to the richness of the function class that the features can represent. Note that we here only state the results under some mild and basic assumptions from the random features literature, and are by no means tight. They can be improved in various ways, for example, if the sampling distribution of $\theta$, $p$, can be data-dependent, and some stronger assumptions on the data and kernel function classes [41, 29].

**Corollary D.2.** Suppose the assumptions in Theorem D.1 hold, and additionally the kernel induced function space $\mathcal{H}$ is rich enough such that $\inf_{f\in\mathcal{H}} \mathcal{E}(f) = 0$. Then, with horizon length $H = \widetilde{\Theta}(\frac{\log(R_{\max}/\epsilon)}{1-\gamma})$ and $M = \widetilde{\Theta}(\frac{1}{(1-\gamma)^3\epsilon^4})$ episodes in dataset $\mathcal{D}$, and with $K = \widetilde{\Omega}((1-\gamma)^{-2}\epsilon^{-2})$ random features, we have $\|\widehat{Q}_\pi^H(w^*) - Q_\pi\|_\infty \leq \mathcal{O}(\epsilon)$ for any $\pi$, where for each $(s,a)$, $\widehat{Q}_\pi^H(s, a; w^*)$ is a $H$-horizon truncation of (D.1), which can be estimated from the offline dataset $\mathcal{D}$.

The proof of Corollary D.2 can be found in §D.2, which specifies the number of random features, the horizon length per episode, and the number of episodes, in order to approximate $Q_\pi$ accurately by using the data in $\mathcal{D}$. Note that the number of random features is not excessive and is polynomial in problem parameters. Combining Theorem D.1 and Corollary D.2 leads to the informal statement in Theorem 3.1.

Next, we justify the use of multi-action $Q$-functions in planning in a deterministic transition environment, which contains all the environments our empirical results we have evaluated. Recall that for any given reward $R$, let $Q_\pi$ denote the $Q$-function under policy $\pi$. Note that with a slight abuse of notation, $Q_\pi$ can also be the multi-step $Q$-function (see definition in §2), and the meaning should be clear from the input, i.e., whether it is $Q_\pi(s,a)$ or $Q_\pi(s, a_1, \cdots, a_H)$.

**Theorem D.3.** Let $\Pi$ be some subclass of Markov stationary policies, i.e., for any $\pi \in \Pi$, $\pi : \mathcal{S} \to \Delta(\mathcal{A})$. Suppose the transition dynamics $\mathcal{T}$ is deterministic. For any given reward $R$, denote the $H$-step policy obtained from $H$-step open-loop policy improvement over $\Pi$ as $\pi_H' : \mathcal{S} \to \mathcal{A}^H$,

defined as

$$\pi'_H(s) \in \underset{(a_1,\cdots,a_H)\in\mathcal{A}^H}{\operatorname{argmax}} \underset{\pi\in\Pi}{\max} Q_\pi(s, a_1, \cdots, a_H),$$

for all $s \in \mathcal{S}$. Finally, define the value-function under $\pi'_H$ as $V_{\pi'_H}(s) := Q_{\pi'_H}(s, \pi'_H(s))$, where $Q_{\pi'_H}(s, a_{1:H})$ is the fixed-point of the Bellman operator $\mathcal{T}_{H,\pi'_H}$ defined in (D.8). Then, we have that for all $s \in \mathcal{S}$

$$V_{\pi'_H}(s) \geq \max_{a_{1:H}} \max_{\pi\in\Pi} Q_\pi(s, a_1, \cdots, a_H) \geq \max_a \max_{\pi\in\Pi} Q_\pi(s, a).$$

## D.2  Deferred Proofs

### D.2.1  Proof of Theorem D.1

The proof relies on the result of generalization guarantees of learning from random features, with squared loss. Note that one cannot use the results in [40], which dealt with Lipschitz loss function of the form $c(y', y) = c(y'y)$. This does not include the squared loss we used in our experiments. Instead, we resort to the results in [41], which also yield a better statistical rate. For the sake of completeness, we re-state the abridged version of one key result therein, Theorem 1 in [41], as follows.

**Lemma D.4.** Suppose that $K$ is a kernel with an integral representation $K(x, x') = \int_\Omega \psi(x, w)\psi(x', w)dp(w)$, where $(\Omega, p)$ is a probability space and $\psi : X \times \Omega \to \mathbb{R}$, where $X$ is a separable space. Suppose $\psi$ is continuous and $|\psi(x, w)| \leq \kappa$ with $\kappa \in [1, +\infty)$ almost surely, and $|y| \leq b$ almost surely. Define the expected risk:

$$\mathcal{E}(f) := \int (f(x) - y)^2 d\rho(x, y),$$

where $\rho$ is the distribution where the data samples $(x_i, y_i)_{i=1}^n$. Define the solution to kernel ridge regression with $M$ random features as

$$\widehat{f}_{\lambda,M}(x) = \phi_M(x)^\top \widehat{w}_{\lambda,M}, \quad \text{with } \widehat{w}_{\lambda,M} := (\widehat{S}_M^\top \widehat{S}_M + \lambda \boldsymbol{I})^{-1}\widehat{S}_M^\top \widehat{y}, \tag{D.2}$$

where $\phi_M(x) := \big(\psi(x, w_1), \psi(x, w_2), \cdots, \psi(x, w_M)\big)/\sqrt{M}$, $w_i$ are drawn i.i.d. from $p(\cdot)$, $\widehat{y} := (y_1, \cdots, y_n)/n^{1/2}$, $\widehat{S}_M^\top := \big(\phi_M(x_1), \cdots, \phi_M(x_n)\big)/n^{1/2}$. Then, suppose $n \geq n_0$, $\lambda = 1/n^{1/2}$, and the number of features $M \geq c_0 \sqrt{n} \log(108\kappa^2\sqrt{n}/\delta)$, we have that with probability at least $1 - \delta$,

$$\mathcal{E}(\widehat{f}_{\lambda,M}) - \min_{f\in\mathcal{H}} \mathcal{E}(f) \leq \frac{c_1 \log^2(18/\delta)}{\sqrt{n}},$$

where $n_0, c_0, c_1$ are absolute constants, $\mathcal{H}$ is the reproducing kernel Hilbert space corresponding to the kernel $K$.

We then apply Lemma D.4, with $(x, y)$ in the lemma being replaced by $\big((s, a), r(s, a)\big)$, $\rho(x, y), p, x, w, M, \lambda$ in the lemma being replaced by $\rho(s, a) \cdot R_{s,a}, p, (s, a), \theta, K, \lambda$ in our case. Note that Lemma D.4 requires the space $X$ to be separable, and our finite space $\mathcal{S} \times \mathcal{A}$ satisfies; it requires $|y|$ bounded, and our reward is absolutely bounded by $R_{\max}$, and thus also satisfies. We hence obtain that with probability at least $1 - \delta$, if the number of random features $K \geq \Omega(\sqrt{n}\log(\sqrt{n}))$, with $n := HM$, then

$$\mathbb{E}_{(s,a)\sim\rho(\cdot,\cdot), r\sim R_{s,a}(\cdot)} \left(r - \sum_{k\in[K]} w_k^* \phi(s, a; \theta_k)\right)^2 \leq \inf_{f\in\mathcal{H}} \mathcal{E}(f) + \mathcal{O}\left(\frac{\log(1/\delta)}{\sqrt{n}}\right) \tag{D.3}$$

where we note that $w^* = (w_1^*, \cdots, w_K^*)$ is the solution to (3.1), and the $\mathcal{E}(f)$ here is defined in Theorem D.1.

For any policy $\pi$ for the MDP, let $Q_\pi$ denote the $Q$-function under policy $\pi$ and the actual reward function distribution $R$, and $\widehat{Q}_\pi(w^*)$ denote the $Q$-function under the estimated reward using random

features:

$$\widehat{Q}_\pi(s, a; w^*) := \mathbb{E}\left[\sum_{h=1}^\infty \gamma^{h-1}\widehat{r}(s_h, a_h; w^*) \,\middle|\, s_1 = s, a_1 = a\right],$$

$$\text{where} \quad \widehat{r}(s, a; w^*) := \sum_{k \in [K]} w_k^* \phi(s, a; \theta_k). \tag{D.4}$$

By Bellman equation, we have that for each $(s, a)$

$$\left|Q_\pi(s, a) - \widehat{Q}_\pi(s, a; w^*)\right| = \left|\int r\, dR_{s,a}(r) + \gamma \sum_{s', a'} Q_\pi(s, a)\mathcal{T}(s' \mid s, a)\pi(a' \mid s')\right.$$

$$\left. - \widehat{r}(s, a; w^*) - \gamma \sum_{s', a'} \widehat{Q}_\pi(s, a; w^*)\mathcal{T}(s' \mid s, a)\pi(a' \mid s')\right|$$

$$\leq \left|\int r\, dR_{s,a}(r) - \widehat{r}(s, a; w^*)\right| + \gamma \cdot \left\|Q_\pi - \widehat{Q}_\pi(w^*)\right\|_\infty.$$

Taking $\sup$ over $s, a$ and organizing the terms, we have

$$\left\|Q_\pi - \widehat{Q}_\pi(w^*)\right\|_\infty \leq \frac{1}{1 - \gamma} \cdot \sup_{s,a} \left|\int r\, dR_{s,a}(r) - \widehat{r}(s, a; w^*)\right|$$

$$\leq \frac{1}{1 - \gamma} \cdot \sqrt{\sum_{s,a} \left(\int r\, dR_{s,a}(r) - \widehat{r}(s, a; w^*)\right)^2} \tag{D.5}$$

$$\leq \frac{1}{1 - \gamma} \cdot \sqrt{\frac{1}{\underline{\rho}}\sum_{s,a} \left(\int r\, dR_{s,a}(r) - \widehat{r}(s, a; w^*)\right)^2 \rho(s, a)} \tag{D.6}$$

$$= \frac{1}{(1 - \gamma)\sqrt{\underline{\rho}}} \cdot \sqrt{\mathbb{E}_{(s,a)\sim\rho(s,a)}\left(\int r\, dR_{s,a}(r) - \widehat{r}(s, a; w^*)\right)^2} \tag{D.7}$$

where (D.5) uses that $\|\cdot\|_\infty \leq \|\cdot\|_2$ for finite-dimensional vectors, (D.6) uses the definition of $\underline{\rho}$. Further, by Jensen's inequality, for each $(s, a)$

$$\left(\int r\, dR_{s,a}(r) - \widehat{r}(s, a; w^*)\right)^2 = \left(\mathbb{E}_{r\sim R_{s,a}(\cdot)}\left[r - \widehat{r}(s, a; w^*)\right]\right)^2 \leq \mathbb{E}_{r\sim R_{s,a}(\cdot)}\left[r - \widehat{r}(s, a; w^*)\right]^2,$$

which, combined with (D.7) and (D.3), gives that

$$\left\|Q_\pi - \widehat{Q}_\pi(w^*)\right\|_\infty \leq \frac{1}{(1 - \gamma)\sqrt{\underline{\rho}}} \cdot \sqrt{\mathbb{E}_{(s,a)\sim\rho(s,a), r\sim R_{s,a}(\cdot)}\left(\int r\, dR_{s,a}(r) - \widehat{r}(s, a; w^*)\right)^2}$$

$$\leq \frac{1}{(1 - \gamma)\sqrt{\underline{\rho}}} \cdot \sqrt{\inf_{f\in\mathcal{H}} \mathcal{E}(f) + \mathcal{O}\left(\frac{\log(1/\delta)}{\sqrt{n}}\right)},$$

which completes the proof. $\qquad\square$

### D.2.2 Proof of Corollary D.2

First, note that with $H = \Theta\left(\frac{\log(R_{\max}/(\epsilon(1-\gamma)))}{1-\gamma}\right)$ ensures that $\|\widehat{Q}_\pi^H(w^*) - \widehat{Q}_\pi(w^*)\|_\infty \leq \mathcal{O}(\epsilon)$, which can be obtained by the boundedness of $r(s, a)$ by $R_{\max}$, and the fact that

$$\gamma^H \frac{R_{\max}}{1 - \gamma} = (1 - (1 - \gamma))^{\frac{1}{1-\gamma}\cdot H(1-\gamma)}\frac{R_{\max}}{1 - \gamma} \leq \left(\frac{1}{e}\right)^{\log(R_{\max}/(\epsilon(1-\gamma)))}\frac{R_{\max}}{1 - \gamma} = \epsilon.$$

Furthermore, since Theorem D.1 requires $n = HM = \widetilde{\Theta}\left(\frac{1}{(1-\gamma)^4\epsilon^4}\right)$, to make sure $\|\widehat{Q}_\pi(w^*) - Q_\pi\|_\infty \leq \epsilon$. Combining these facts yields the desired result. $\qquad\square$

### D.2.3 Proof of Theorem 3.2

Define

$$Q_H^{\max}(s, a_1, \cdots, a_H) := \max_{\pi \in \Pi} Q_\pi(s, a_1, \cdots, a_H), \text{ and } Q^{\max}(s, a) := \max_{\pi \in \Pi} Q_\pi(s, a).$$

We also define the Bellman operator under the open-loop policy $\pi'_H$ as follows: for any $Q \in \mathbb{R}^{|\mathcal{S}| \times |\mathcal{A}^H|}$

$$\mathcal{T}_{H, \pi'_H}(Q)(s, a_1, \cdots, a_H) = \mathbb{E}\left[ \sum_{h \in [H]} \gamma^{h-1} r(s_h, a_h) + \gamma^H Q(s_{H+1}, \pi'_H(s_{H+1})) \,\Big|\, s_1 = s, a_{1:H} \right].$$
(D.8)

Note that $\mathcal{T}_{H, \pi'_H}$ is a contracting operator, and we denote the fixed point of the operator as $Q_{H, \pi'_H} \in \mathbb{R}^{|\mathcal{S}| \times |\mathcal{A}^H|}$, which is the $Q$-value function under open-loop policy $\pi'_H$. By definition, we also know that the state-value function under $\pi'_H$, $V_{H, \pi'_H} = Q_{H, \pi'_H}(s, \pi'_H(s))$, i.e., by applying the open-loop policy $\pi'_H$ to the actions $a_{1:H}$ in $Q_{H, \pi'_H}(s, a_{1:H})$.

Note that

$$\mathcal{T}_{H, \pi'_H}(Q_H^{\max})(s, a_1, \cdots, a_H) = \mathbb{E}\left[ \sum_{h \in [H]} \gamma^{h-1} r(s_h, a_h) + \gamma^H Q_H^{\max}(s_{H+1}, \pi'_H(s_{H+1})) \,\Big|\, s_1 = s, a_{1:H} \right]$$

$$= \mathbb{E}\left[ \sum_{h \in [H]} \gamma^{h-1} r(s_h, a_h) + \gamma^H \max_{a_{H+1:2H}} Q_H^{\max}(s_{H+1}, a_{H+1:2H}) \,\Big|\, s_1 = s, a_{1:H} \right] \quad \text{(D.9)}$$

$$\geq \mathbb{E}\left[ \sum_{h \in [H]} \gamma^{h-1} r(s_h, a_h) + \gamma^H \max_{a_{H+1:2H}} Q_\pi(s_{H+1}, a_{H+1:2H}) \,\Big|\, s_1 = s, a_{1:H} \right] \quad \text{(D.10)}$$

$$\geq \mathbb{E}\left[ \sum_{h \in [H]} \gamma^{h-1} r(s_h, a_h) + \gamma^H Q_\pi(s_{H+1}, \pi(s_{H+1}) \cdots, \pi(s_{2H})) \,\Big|\, s_1 = s, a_{1:H} \right], \quad \text{(D.11)}$$

$$= Q_\pi(s, a_1, \cdots, a_H), \quad \text{(D.12)}$$

for any $\pi \in \Pi$, where (D.9) uses the definition of $\pi'_H$, (D.10) is due to the definition of $Q_H^{\max}$, and (D.11) is by the $\max_{a_{H+1:2H}}$, and (D.12) is by definition. Since (D.12) holds for any $\pi \in \Pi$, by the monotonicity of $\mathcal{T}_{H, \pi'_H}$, we have

$$Q_{H, \pi'_H}(s, a_{1:H}) = \lim_{k \to \infty} (\mathcal{T}_{H, \pi'_H})^k (Q_H^{\max})(s, a_{1:H}) \geq Q_H^{\max}(s, a_{1:H}) \geq \max_{\pi \in \Pi} Q_\pi(s, a_{1:H}).$$
(D.13)

Notice that for all $s$, by applying $\pi'_H(s)$ on both sides of (D.13),

$$V_{H, \pi'_H}(s) = Q_{H, \pi'_H}(s, \pi'_H(s)) \geq \max_{\pi \in \Pi} Q_\pi(s, \pi'_H(s)) = \max_{a_{1:H}} \max_{\pi \in \Pi} Q_\pi(s, a_{1:H}). \quad \text{(D.14)}$$

Further, due to the multi-step maximization, we have

$$\max_{a_{1:H}} \max_{\pi \in \Pi} Q_\pi(s, a_1, \cdots, a_H) \geq \max_a \max_{\pi \in \Pi} Q_\pi(s, a),$$

which, combined with (D.14), completes the proof. $\qquad\square$

## E   Complexity Analysis

The space complexity of RaMP is primarily determined by the number of random features that are needed. As we describe in Corollary D.2, we require $K = \widetilde{\Omega}((1 - \gamma)^{-2} \epsilon^{-2})$ random features to achieve $\epsilon$-order error between the estimated and true Q-function. As the required approximation error decreases, the space needed to store all the features grows sublinearly. Note that this result is for *any* given reward function (including the target one) under *any* policy $\pi$, but not tied to a specific one (under the realizability assumption of the reward function).

On the other hand, the space complexity of the classical successor feature method is relatively *fixed* in the above sense: the dimension of the feature is fixed and does not change with the accuracy of approximating the optimal Q-function for the target task. However, the resulting guarantee is also more restricted: it is only for the optimal Q-function of the specific target reward, and also depends on the distance between the previously seen and the target reward functions. Hence, the two space complexities are not necessarily comparable.

# F   Relationship to Existing Work

We briefly connect our proposed algorithm to prior work.

**Successor features for transfer in RL:**   While successor features [5, 4] have shown the ability to transfer across problems in RL, the two key differences in our framework are (1) using random features rather than requiring learned or pre-provided features and (2) training multi-action Q functions rather than typical $Q_\pi$. These two changes allow transfer to happen across a broader class of reward functions and not simply be restricted to the policy cover experienced at training time.

**Model-based RL:**   Our work is connected to model based RL in that it disentangles dynamics and rewards, but is crucially different in that it doesn't model one-step evolution of state but rather long term accumulation of random features. This trades off compounding error for generalization error.

**Model-free RL:**   Our work is connected to methods for model-free RL in that it also models a set of Q-functions, but importantly this doesn't correspond to a particular reward, but rather to random features of state. By doing so, we are able to adapt to arbitrary rewards at test-time, rather than being tied to a particular reward function.

