# OpenReview forum: "Self-Supervised Reinforcement Learning that Transfers using Random Features"
_NeurIPS.cc/2023/Conference — NeurIPS 2023 poster_

### Official Review · Reviewer_8qBK · 2023-06-15

**Soundness:** 3 good
**Presentation:** 2 fair
**Contribution:** 3 good
**Rating:** 5
**Confidence:** 4

**Summary:**

The authors propose a self-supervised method to learn approximate multi-step Q-estimates by leveraging random features as bases for a reward function in the target task. This approach addresses some limitations of model-based and model-free RL. In particular, model-based RL typically suffers from compounding error of predicting future states from predicted states. On the other hand, model-free RL tends to suffer from instability when transferring knowledge. The proposed method mitigates these problems by implicitly learning transition dynamics (without explicitly modeling future trajectories) from offline data in a self-supervised manner based on the prediction of random features.

**Strengths:**

The methodology is interesting. Using random features in a self-supervised manner to form a Q-basis such that one can find a linear combination of these random features for the reward function of a downstream task using approximate multi-step returns makes sense and seems novel.

**Weaknesses:**

The main weakness of this approach lies in the experiments and poor framing of how this work fits in the context of meta-RL. meta-RL is not mentioned, but it entails this same problem of having multiple tasks with shared dynamics such that the reward function changes from one task to another. As such, the authors should be comparing against meta-RL benchmarks, not standard RL. It is interesting because the authors are running their experiments on a popular meta-RL benchmark called Meta-World and yet they do not compare their approach against meta-RL approaches. One of the reasons why this is currently an unfair comparison is the need for the competing approaches to learn their value functions, etc. from scratch, whereas the proposed approach had the benefit of learning from an offline dataset befrorehand and adapting to the target task.

**Questions:**

1. I am confused on your statement that the Q-function for multi-step returns is independent of the policy when H = \tauThe lim. You still have \tilde{a}_1, \tilde{a}_2, ..., \tilde{a}_\tau. From what policy do these actions come from?

2. Why do you not compare against meta-RL methods? The successor features approach you compare against may count as meta-RL (it is not called that in their original paper though), but it is a method from 2017.

**Limitations:**

The limitations have been addressed.

---

> ### Author Rebuttal · Authors · 2023-08-10
>
> We thank the reviewer for their feedback. We respond to your comments below.
>
> > The main weakness of this approach lies in the experiments and poor framing of how this work fits in the context of meta-RL. meta-RL is not mentioned, … beforehand and adapting to the target task.
>
> Thanks for the suggestion. We did not compare with meta-RL as usually *rewards* are needed for this case (in both the meta-training and meta-testing phases), while using a self-supervised approach without requiring reward labels is exactly one of the advantages of our approaches. By not requiring reward, our method avoids assumptions about train/test time reward sharing the same distribution, as is typical in meta-RL. Given this problem setting of having unlabelled offline datasets and distribution shift between meta-train and meta-test, comparing these settings is not quite apples to apples. However, as per your suggestion, we have added new experimental results and comparison with meta-RL algorithm RL2 in figure 1 of rebuttal pdf. We note that since our problem setting isn't typical meta-rl, which assumes the train/test time objectives are in the same underlying distribution, we have to adopt RL2 by giving it privileged information similar to our adoption of CQL. Out of the four environments which we ablate RL2, it's similar to our method in door open but inferior in all others. This shows that meta-rl will suffer from out-of-distribution goals which our method avoids.
>
> > I am confused on your statement that the Q-function for multi-step returns is independent of the policy when $H = \tau$ The lim. You still have $\tilde{a}_1, \tilde{a}_2, ..., \tilde{a}_\tau$. From what policy do these actions come from?
>
> Great question! In a typical Q function $Q^\pi(s, a)$ refers to expected value when a is taken at s, and the policy \pi is followed thereafter. This makes the dependence on the policy $\pi$ implicit, and the same Q-function cannot be used to evaluate a different policy. In a multi-step open-loop Q-function $Q(s, \tilde{a}_1, \tilde{a}_2, ..., \tilde_{a})$, the dependence on *actions* is *explicit*, i.e., the Q-value, is completely *determined* directly by this sequence of actions (and does not have any implicit dependence on any specific policy $\pi$). When trying to find the optimal sequence of actions, different sequences of actions are generated randomly and then evaluated using the learned Q-function to choose the *best* sequence of actions. Essentially, any policy that can generate the sequences of actions with a good coverage of all possible sequences would be sufficient. In doing so, the same multi-action Q function can evaluate many different action sequences within the data, without having any implicit policy dependence like standard Q-learning. An implicit policy dependence would make it hard to use the same Q function to evaluate many different action sequences.
>
> Please feel free to let us know if there are any other comments that may help reevaluate our paper.

---

> > ### Comment · Reviewer_8qBK · 2023-08-18
> > **Updated experiments**
> >
> > Thank you for the updated experiments on the pdf. I have raised my score. Should the paper be accepted, please include these results and explanation of meta-RL and how you differ.

---

> ### Author Response · Authors · 2023-08-15
>
> Thank you for the helpful suggestions! If you have additional questions or experiment suggestions regarding our rebuttal, we are happy to answer that!

---

### Official Review · Reviewer_BJ94 · 2023-06-25

**Soundness:** 2 fair
**Presentation:** 3 good
**Contribution:** 3 good
**Rating:** 5
**Confidence:** 4

**Summary:**

The paper introduces RaMP, an approach for fast adaptation to unseen reward functions given offline data collected with arbitrary behavior policies under the same dynamics. RaMP learns a set of basis multi-step Q-functions, each corresponding to a random reward defined as the accumulation of random state action features. For online adaption to new rewards, the new reward/Q function is then identified using linear regression given the basis functions and used for control via MPC.

**Edit After Rebuttal**:

Raised score from 3 to 5, see comment below for details

**Strengths:**

RaMPs' key idea, learning "random feature rewards" which are later combined, is novel and interesting as it allows offline pertaining without "reward labels" and provides an easy and efficient adaptation mechanism. The paper addresses a significant problem, the generalization to novel rewards under little to no assumptions on the knowledge of the current scenario and previously seen rewards, and is mostly clearly written and easy to follow.

**Weaknesses:**

The paper's main weakness is its experimental evaluation, which I think is in its current form insufficient to validate the paper's claims or allow assessment of the method's potential for several reasons:

- The algorithm described in the main paper seems to be improved by several "implementation details" (c.f. Appendix B). While the paper states that "RaMP’s performance already surpasses the baselines even without them", I believe a rigorous analysis of their impact would be necessary to assess the method's potential and workings.
-  Baselines: none of the considered model-based RL methods (PeTS, MBPO, Dreamer) was designed to work with dynamics models trained on **offline** data. For a fairer comparison, methods designed to work with such dynamics models (e.g.  MOPO[1]) should be considered.
- While RaMP adapts very fast in the considered environments, its final performance is often lower than that of MBPO and/or PeTS, in particular for the Hopper and D'Claw experiments. Further discussion and analysis in the paper would allow a better understanding of RaMPs limitations.
- Compared with most recent work in RL and/or offline RL the paper only uses a small set (8) of relatively simple environments - the maximal state dimension is 39 (meta world, where only a subset of these is relevant for the considered tasks) and maximal action dimension is 9.
- Higher dimensional observations (pixels) are used for some experiments in the supplement but the results are again inconclusive or seem to favor the baseline (Dreamer-V1, which is to the best of my knowledge not SOTA on pixel metaworld [2])
- See questions below.

While the majority of the paper is clearly written and understandable, this is, in my opinion, not the case for section 3.4. Here the assumptions, their realisticness, and practical consequences of theorem 3.1 could be stated much more clearly

[1] MOPO: Model-based Offline Policy Optimization, Yu et al 2020

[2] Masked World Models for Visual Control, Seo et al 2022


**Questions:**

- I am a bit puzzled by the meta-world return curves: If I recall correctly successful execution in most of these tasks corresponds to an episode return of ~ 4,000. However, all presented algorithms achieve max. 1,000 (on "Reach wall") and considerably less in the 3 other tasks. Is there some normalization here? I would prefer to (additionally) see the success rates for these tasks to improve the interpretability of the results.
- What exactly is indicated by the error bars? How statistically significant are these results given that it's only 4 seeds?
- Are the "implementation details" (Appendix B) also used for the baselines? In particular: Additional value function for infinite horizon (B.1) for PeTS, online adaptation (B.2, but adapting the dynamics) for PeTS and MBPO, Planning With Uncertainty (B.3) for PeTS and MBPO, and MPPI (B.4) for PeTS.

- What are the standard "coverage and sampling assumptions" in theorem 3.1? How realistic are those in praxis? What does the theorem add, compared with standard random feature regression results?

**Limitations:**

I believe all limitations have been addressed, although some only in the supplement (infinite horizon problematic with multi-step q function), and I believe the paper would benefit from moving this also to the main part.

---

> ### Author Rebuttal · Authors · 2023-08-10
>
> We thank the reviewer for their comments.
>
> > The algorithm described in the main paper seems to be improved by several "implementation details" ... I believe a rigorous analysis of their impact would be necessary to assess the method's potential and workings.
>
> Thanks for the suggestion. We will move more details from Appendix B back to the main paper, and make the statement more rigorously by  provide ablation experiments addition to those in appendix to validate the importance of each design choice.
>
> > Baselines: none of the considered model-based RL methods (PeTS, MBPO, Dreamer) was designed to work with dynamics models trained on offline data. For a fairer comparison, methods designed to work with such dynamics models (e.g. MOPO[1]) should be considered.
>
> Thanks for bringing up MOPO. We’d like to clarify that  MOPO isn’t directly applicable here while also providing results of adopted MOPO by providing it with privileged information.  MOPO adds a penalization term to reward based on ensemble disagreement during policy training while our offline training data contains NO reward. Despite this, we adopt MOPO in a way similar to how we adopt CQL in figure 1 in rebuttal pdf. Despite privileged information, we observed that MOPO will suffer from similar problems that goal-conditioned CQL faces.
>
> > While RaMP adapts very fast in the considered environments, its final performance is often lower than that of MBPO and/or PeTS ... Further discussion and analysis in the paper would allow a better understanding of RaMPs limitations.
>
> Thank you for the suggestion. We will add relevant discussion and analysis about this behavior in our final version. These differences in the 3 environments are due to the intrinsic problem of random-shooting MPC compared to policy based-methods. We recognize this limitation and would address accordingly in final version
>
> > Compared with most recent work in RL and/or offline RL the paper only uses a small set (8) of relatively simple environments ... maximal action dimension is 9.
>
> If appendix is included, our paper has a total of 15 environments before the rebuttal. We also just added some D4RL results in figure 4 of rebuttal pdf. We note that we are NOT under an offline-reinforcement learning setup so suitable benchmarks aren't directly applicable here. We note a lot of highly cited works under similar setting like successor features only has 2 environments with much lower action dimension.
>
> > Higher dimensional observations (pixels) are used for some experiments in the supplement but the results are again inconclusive or seem to favor the baseline (Dreamer-V1, which is to the best of my knowledge not SOTA on pixel metaworld [2])
>
> We present the pixel observation result mainly to illustrate that random projection also works on higher dimensional data and convolution networks rather than presenting our method as the state-of-the-art in RL from pixels. We will revise the writing and clarify the purpose of the experiment there.
>
> > If I recall correctly successful execution in most of these tasks corresponds to an episode return of ~ 4,000. However, all presented algorithms achieve max. 1,000 (on "Reach wall") and considerably less in the 3 other tasks.
>
> Metaworld is an environment which doesn’t explicitly provide a “done” signal for the environment. It didn’t provide a max episode length either so the benchmark with metaworld needs a manually specified max episode length. We chose a short episode length here since the four metaworld tasks don't need excessively many steps as in the original paper.
>
> > What exactly is indicated by the error bars? How statistically significant are these results given that it's only 4 seeds?
>
> The error bars here are standard errors. We deem 4 seeds to be statistically significant since, unlike online reinforcement learning, we have a static dataset which majority of the training steps are trained on. This makes the models already very stable before the online adaptation phase and also avoids the variance from blind exploration.
>
> > Are the "implementation details" (Appendix B) also used for the baselines? In particular: Additional value function for infinite horizon (B.1) for PeTS, online adaptation (B.2, but adapting the dynamics) for PeTS and MBPO, Planning With Uncertainty (B.3) for PeTS and MBPO, and MPPI (B.4) for PeTS.
>
> Yes, all baseline results are with online adaptation. For Planning With Uncertainty, we have a model ensemble for MBPO and SF baseline and a regularized version, MOPO in rebuttal pdf figure 1.
>
> > What are the standard "coverage and sampling assumptions" in theorem 3.1? How realistic are those in praxis? What does the theorem add, compared with standard random feature regression results?
>
> We have stated in the theorem that it is an “informal” version of the full result, due to space constraints. We will move the full version back to the main paper. The coverage and sampling assumptions, as we stated formally in Theorem D.1, means that the state-action pair is sampled from some offline data distribution that has coverage of the state-action space (diverse enough), and this is related to the standard “all policy concentrability” assumption in the offline RL theory literature, e.g., [1,2]. It is necessary for many offline RL approaches [3], and is important for our approach to be able to cover *any* new reward functions at test time. Compared to the random feature literature, we additionally need to analyze the propagation of the reward function approximation error to value function approximation (due to sequential decision-making), to adapt the loss used in the literature to that in our setting, and to analyze the effect of multi-step optimization over the Q-functions. We will make these points more explicit.
>
>
> [1] Finite-time bounds for fitted value iteration
>
> [2] Approximate policy iteration schemes: A comparison
>
> [3] Information-theoretic considerations in batch reinforcement learning

---

> > ### Comment · Reviewer_BJ94 · 2023-08-14
> >
> > The authors thoroughly addressed my concerns and provided additional results regarding MOPO and the “implementation details” Given these and the promised revision, I’ll increase my score.
> >
> > I still would urge the authors to reconsider the metaworld results though. Why the official codebase is indeed ambiguous in this regard I believe it is common practice to run the tasks for 500 steps each (until an exception is thrown) and report the success rate based on the indicator in the *info* dictionary. Following this protocol would greatly improve the compatibility (with other, potential future, works) and interpretability of these results.

---

> ### Author Response · Authors · 2023-08-15
>
> Thank you for the helpful suggestions! We will try to provide such data or explanation in the final version.

---

### Official Review · Reviewer_1b4X · 2023-06-28

**Soundness:** 4 excellent
**Presentation:** 3 good
**Contribution:** 3 good
**Rating:** 7
**Confidence:** 4

**Summary:**

The paper proposes an approach for unsupervised pre-training of RL agents, ie pre-training on offline agent experience without rewards. The approach generates a set of random reward functions and then learns a successor representation of the state-action space for predicting cumulatives of these rewards on fixed-horizon trajectories from the pre-training data. At test time, it uses a small reward-annotated dataset to learn a linear mapping from the pre-trained successor representation to the target reward. Then it extracts a policy by greedily maximizing the resulting estimates of future cumulative rewards. The paper demonstrates that this allows for faster finetuning than model-based RL since only a linear mapping needs to be learned on the target task data instead of the full Q-function. In contrast to prior work on successor features it does not assume access to given features or a reward-annotated training set to learn features, but instead uses random reward projections.

**Strengths:**

- the problem formulation is impactful: pre-training with reward-free data has great potential for generalist RL agents

- the paper is well-written and easy to follow, it clearly outlines the problem formulation, explains the method and provides theoretical justifications

- the experimental evaluation seems comprehensive, with only minor experiments missing (see below): the paper compares to a representative set of baselines on multiple environments, including image-based control (though there with limited success) and also performs ablations of the key elements of the method



**Weaknesses:**

(A) **Relation to prior work not sufficiently explained**: the proposed approach heavily builds on top of prior work on success features for RL, yet the current submission only mentions this relation in a half-paragraph in the main paper and adds a slightly more detailed discussion at the very end of the appendix. It would be helpful to more clearly introduce this most relevant prior work and explain the deltas, so it is easier for readers to understand the main novel components. Concretely, one could add a preliminaries section that summarizes the idea of successor features and then point out the two main novelties in the proposed approach: random reward projections to be independent of training tasks and action chunking to be independent of training policies.

(B) **Random reward features don't scale well to images**: the main difference to prior successor feature works is that the proposed approach avoids the assumption of pre-defined features or given pre-training task distributions by using random reward projections. The downside of this is that the algorithm cannot "discard" any information and struggles on high-dimensional inputs like images, where predicting random rewards over pixel inputs is complex and lacks semantic meaning. I agree with the authors that this can potentially be mitigated with pre-trained representations, but since removing the assumption of pre-defined features is one of the main selling points of this work, this tradeoff should be discussed more prominently in the main paper (currently only in appendix C.3) and the corresponding image-based results should be included in the main text. It would also be helpful to show the successor feature baseline results on the image-based domain.

(C) **Missing Ablation**: The other introduced novelty is the use of action chunking, ie conditioning the Q-function on a sequence of H actions instead of on a single action. However, this choice is not ablated. It would be good to include versions of the proposed approach without action chunking (ie H = 1) and with different horizon lengths to see the benefit of chunking.

(D) **Missing baseline**: The paper compares to goal-conditioned offline RL (CQL) but with a pre-specified selection of goals. Prior work has instead proposed to use goal-conditioned offline RL on randomly sampled goal states from an offline dataset for pre-training [1]. This has two benefits: (1) it does not require pre-defined goals / tasks, fully matching the assumptions of the proposed approach, (2) because of that it may be more robust to test time adaptation to new tasks. Thus, it would be good to add comparison to Actionable Models on all environments.

**Minor Point**:

(E) **MPC is misnomer**: The paper refers to the downstream policy extraction as "model predictive control" and "planning" (see eg Fig 2). However it does not perform prediction or use a model. Instead it performs greedy policy extraction by maximizing the Q-function, but uses action chunking (ie N-step action inputs). Thus I would call this step "greedy policy extraction with action chunking" instead.

[1] Actionable Models, Chebotar et al. 2021


## Summary of Review

Overall I like the paper and am happy to accept it if the authors can more clearly mention the connections to prior work on successor features, discuss tradeoffs for image-based domains and add the suggested ablation + baselines. The idea of using random reward projections for successor feature learning is novel to my knowledge, but I am not 100% certain and will thus assign lower confidence to my review.




**Questions:**

N/A

**Limitations:**

Limitations on image-based environments are demonstrated in the appendix but should be discussed more prominently in the main paper (see weaknesses (B)).

---

> ### Author Rebuttal · Authors · 2023-08-10
>
> We thank the reviewer for their comments. We respond to your comments below.
>
> > Relation to prior work not sufficiently explained.
>
> Thank you for the great suggestion. We will add a preliminary section to summarize the idea of successor features, and emphasize our novelties in the section, in our final version of the paper. This would indeed make our contributions easier and more outstanding for readers to understand. You’ve hit the nail on the head with the key elements - random features rather than require pre-known successor features, and getting rid of policy dependence by doing open loop Q modeling wrt action sequences rather than having an implicit policy dependence. These are crucial because it allows us to learn from offline datasets without reward labels, and it allows us to learn Q-functions that transfer significantly better than techniques like generalized policy iteration (GPI). GPI can simply take the piecewise max between many optimal policies, while RaMP can optimize for any policy that is within the data support. We will make this significantly more clear in the document.
>
> > Random reward features don't scale well to images.
>
> Thanks for the insightful comment. While RaMP can work directly from images using random convolutional features, as we have shown in Appendix C.5, this may perform more effectively using pretrained image features rather than using completely random convolutional features.
>
> While one of our motivations is to *remove* the pre-defined features, what we meant by using *pre-trained* features, as a remedy to the scalability issue, is that we will first *randomly* project onto some feature space of the images, e.g., the latent space pre-trained over standard datasets as ImageNet, and then do random MLP features building on these pretrained features (as in our state based experiments). This is a fairly standard approach to reduce the dimension to consider, when dealing with images and still allows there to be random features, but now building on top of pretrained image features (not directly using those image features). We will make this point more explicit, and also move the related discussions and image-based results from the appendix to the main paper.
>
> > Missing Ablation.
>
> Thanks for the suggestion. We ablated the horizon earlier under the finite horizon variant by shortening the horizon by half and four times. We found that longer-horizon environments like hopper would suffer slightly while our method retains similar performance on meta-world environments. Unfortunately, we ran out of time benchmarking enough seeds for this ablation under our current infinite horizon variant. We appreciate the suggestion and will add final results to the paper in the future.
>
> > Missing baseline - Actionable models
>
> Since source code is not available, we were not able to run this baseline in the short rebuttal period but we will certainly run this and include it in the final version. We would like to note that while actionable models are restricted to goal conditioned problems, RaMP can adapt to any reward function, even non goal reaching ones.
>
> > Minor point – MPC is a misnomer.
>
> Thanks for the suggestion. The step uses “prediction” in the sense that the Q-function we used for optimization is an “accumulated” reward over $H$ steps, and it belongs to the MPC framework as we make “optimization” over a window of $H$ steps, but only *implement* the first step, and then interact with the environment before “re-optimize” the sequence of actions again. However, we certainly acknowledge the confusion here and we will rename the method accordingly.
>
> Please feel free to let us know if there are any other questions.

---

> > ### Comment · Reviewer_1b4X · 2023-08-16
> > **Thank you for your rebuttal!**
> >
> > Thanks for providing the rebuttal. I am generally happy with the response and appreciate the authors willingness to incorporate the feedback. I also understand that the short rebuttal window can make it hard to implement new ablations / baselines on the spot, particularly if many seeds are required. I trust that the authors will follow through on their promised changes. Please also add comparison to baselines on image-based domains as mentioned in my review.
> >
> > I have also looked over the reviews of other reviewers specifically to understand the low ratings of reviewers 8qBK and gqpD. After reading through the reviews and rebuttals I find the criticisms rather surface-level and don’t fully agree with them:
> > - compare to meta-RL (R-8qBK): agreed with the authors that meta-RL requires training tasks / rewards, which their method does not. There is work on unsupervised meta-RL though that could allow for clean comparison (Gupta et al 2018, https://arxiv.org/abs/1806.04640)
> > - compare to IQL pre-training (R-gqpD): I don’t think IQL w/ final reward 1 for value pre-training is ideal on non-expert data — the proposed projection after random reward pre-training seems more general
> >
> > I believe the authors adequately addressed the concerns of these reviews so I stand by my accept recommendation and am willing to defend it to the AC. I will also increase my confidence given that no other reviewer raised concerns about too closely related work.

---

> ### Author Response · Authors · 2023-08-15
>
> Thank you for the helpful suggestions! If you have additional questions or experiment suggestions regarding our rebuttal, we are happy to answer that!

---

### Official Review · Reviewer_gqpD · 2023-07-06

**Soundness:** 3 good
**Presentation:** 4 excellent
**Contribution:** 2 fair
**Rating:** 7
**Confidence:** 2

**Summary:**

The paper tackles the problem of

In the training phase, they create $H$ randomly initialized neural networks that serve as reward functions during the purpose of pretraining. The Q-function learned online is a linear combination of these random features: $w^T\sum_{h\in [H]}\phi(s_h,a_h)$.

**Strengths:**

This is a beautifully written paper. The motivation is crisp and the proposed method is very well grounded theoretically. Results across a broad range of tasks are presented. On originality, clarity, and method quality, I rate this paper highly.

**Weaknesses:**

The paper would benefit from more baselines: for example doing something like  IQL with a reward of 1 at the end of each trajectory during offline training. Less importantly, I also think IQL would be a more natural oracle than CQL.  It performs better than D4RL and directly evaluates online fine-tuning. In fact, the IQL paper shows that CQL can sometimes collapse during online fine-tuning.

I also don't quite understand the choice of offline dataset: from the appendix, it sounds like the policies that generate the offline dataset are quite "expert." It would be interesting to see how well the model performs if the offline dataset was instead modified from the D4RL benchmark, which includes different levels of expert policies. I expect high quality data to benefit any self-supervised method, but the claims of the paper are not limited to offline learning on expert demonstrations.

**Questions:**

I'm a little confused by Figure 2. The way that the random features and Q-basis are drawn makes it look like each $\phi_h$ is one of the basis vectors that together compose $Q-basis$. Everywhere in the text implies that each index of Q-basis is a different function parameterized by a different $\theta_k$ and together the $\theta_k$ make up the basis vector. Is that correct? If so I would recommend modifying Figure 2 left to include a reference to k.

The number of environment steps in the online phase seems really small. Why was 3000 chosen and about how many trajectories is it? E.g., in the IQL fine-tuning experiments they use ~1M environment steps.

Why did the authors not include D4RL data in their experimental evaluations?

---

> ### Author Rebuttal · Authors · 2023-08-10
>
> We thank the reviewer for their comments. We respond to your comments below.
>
> > The paper would benefit from more baselines: for example doing something like IQL with a reward of 1 at the end of each trajectory during offline training.
>
> Thanks for the great suggestion. We have added new experimental results by comparing with IQL with the reward scheme that you suggested (see Fig. 1). This is akin to the CQL baseline in the original paper, but replacing the base algorithm CQL with the newer algorithm IQL. As expected, IQL performs less favorably compared to our approach, as CQL did. We will add the new experiments in our final version. The important point to note here is that model-free offline RL methods like CQL/IQL are inherently tied to the reward, while RaMP is not tied to the rewards but effectively models the system dynamics. This allows it to easily transfer to new rewards, in situations where algorithms like CQL/IQL will struggle to transfer since they are inherently tied to the reward.
>
> > I also don't quite understand the choice of offline dataset: from the appendix, it sounds like the policies that generate the offline dataset are quite "expert."
>
> We would like to clarify the dataset setup and present new results on D4RL. We use noised expert policy to collect trajectory with very aggressive epsilon noise (>= 0.5) so the trajectories rarely reach high reward. In addition, we don’t assume the dataset contains rewards so the test time objective can be quite challenging for the neural network to learn. In Fig. 4 of the supplementary pdf, we have added a D4RL experiment where we outperform CQL. The new experiment shows that our comparison isn’t due to how our data is collected. The important point is that the datasets must have coverage, as is typical in most offline RL algorithms, but the datasets can be very mixed with both optimal and suboptimal behavior. This is in contrast to imitation learning datasets that only have expert data.
>
> > It would be interesting to see how well the model performs if the offline dataset was instead modified from the D4RL benchmark, which includes different levels of expert policies.
>
> Thanks for the great suggestion. We have added the use of D4RL in our new experiments, see Fig 4. The original reason why we didn’t benchmark in D4RL is that we want to develop a method that doesn’t assume train/test time objective correlation, and thus want to test on out-of-distribution goals or rewards. Most of the D4RL environments are not goal-conditioned and have the same reward function during the offline/online phase. This is drastically different from our reward-agnostic setting. In our new experiment, we chose only few goal-conditioned environments in D4RL and the results illustrate our performance isn’t due to our customized data but the algorithm itself.
>
> > I'm a little confused by Figure 2. The way that the random features and Q-basis are drawn makes it look like each
> … Is that correct? If so I would recommend modifying Figure 2 left to include a reference to k.
>
> Sorry for the confusion. It is correct that the basis used to estimate Q is different from (but the accumulation of) the basis used to estimate the reward function, which is $\psi$ instead of $\phi$. For the features $\phi$ and $\psi$, their “parameterizations” are different also – the former is parameterized by $\theta$, while the latter is parameterized by $\nu$ (See our Sec. 3.2). We will make the clarifications in our updated version.
>
> > The number of environment steps in the online phase seems really small. Why was 3000 chosen and about how many trajectories is it? E.g., in the IQL fine-tuning experiments they use ~1M environment steps.
>
> One of the main baselines, Successor Features, is extremely slow to run in terms of clock time. The 32000 step is a hyper-parameter which we choose by observing the number of environment steps needed to achieve success/converge across environments. Since planning backbone isn’t our main focus, we used a computationally inefficient version of random shooting MPC for both Successor Features and our method. Random projection for the successor feature baseline using many independent MLPs instead of one big MLP also made it slow, not to mention we have to copy these networks multiple times for ensemble. Official MBPO implementation is also very slow with ensembles so it’s also extremely difficult to train ~1M steps.
>
> Please feel free to let us know if there are any other comments that may help reevaluate our paper.

---

> > ### Comment · Reviewer_gqpD · 2023-08-21
> > **Thank you**
> >
> > Thank you for taking the time to address these concerns. Based on the response, I raised my recommendation to Accept.

---

> ### Author Response · Authors · 2023-08-15
>
> Thank you for the helpful suggestions! If you have additional questions or experiment suggestions regarding our rebuttal, we are happy to answer that!

---

### Official Review · Reviewer_W1dV · 2023-07-31

**Soundness:** 3 good
**Presentation:** 2 fair
**Contribution:** 3 good
**Rating:** 4
**Confidence:** 4

**Summary:**

The paper proposes a Random Features for Model-Free Planning (RaMP) algorithm to solve the problem of learning generalist agents that are able to transfer across platform where the environment dynamics are shared, but reward function is changing. The proposed algorithm leverages diverse unlabeled offline data to learn models of long horizon dynamics behavior, while being able to naturally transfer across tasks with different reward function. The authors evaluated RaMP on number of simulation based robotic manipulation and locomotions tasks.

**Strengths:**

- The problem addressed in the paper is very important problem for RL in robotics/robot learning real-life tasks.
- The proposed method is very interesting and seems relevant to the research community.

**Weaknesses:**

- The paper is not well-written and very hard to follow. For example, Line 13-15 and Line 59-61 is very hard to follow.
- The paper attempts to solve an interesting problem and has strong results in simulation based robotics task. Since, the motivation of the paper draws from real-robotic tasks, it would be good to see how this method performs on real-robotic tasks. I understand authors have mentioned in future work but to understand complete effectiveness of the proposed method, it seems critical.
-

**Questions:**

Please address concerns mentioned in the weakness section.

**Limitations:**

mentioned in the paper.

---

> ### Author Rebuttal · Authors · 2023-08-10
>
> We thank the reviewer for their comments. We respond to your comments below.
>
> > The paper is not well-written and very hard to follow. For example, Line 13-15 and Line 59-61 is very hard to follow.
>
> Thanks for the comment. Line 13-15 simply means our method can be trained “without” reward labels, but on the other hand enjoy the benefit of being able to be quickly deployed to new tasks. Line 59-61 simply means that as long as the number of random features being used is large enough, we can estimate any test-time Q-function by a linear combination of the Q-basis functions. We will improve the writing in our updated version.
>
> >The paper attempts to solve an interesting problem and has strong results in simulation based robotics tasks. Since, the motivation of the paper draws from real-robotic tasks, it would be good to see how this method performs on real-robotic tasks. I understand authors have mentioned in future work but to understand the complete effectiveness of the proposed method, it seems critical.”
>
> Thanks for the great suggestion. We agree that applying our method to real-robotic tasks would be a big plus. With the timeline constraints, we may not be able to finish such real-robotic experiments during the rebuttal phase. We will add them in the next version of our paper. We would like to emphasize that we have run our method across 8 different problem domains, and provide theoretical backing of the proposed algorithm. Hopefully, this provides convincing evidence for the reviewers of the empirical efficacy of our proposed method.

---

> ### Author Response · Authors · 2023-08-15
>
> Thank you for the helpful comments. If you have additional questions or experiment suggestions regarding our rebuttal, we are happy to answer that!

---

### Author Rebuttal · Authors · 2023-08-10

We thank the reviewers for their comments and suggestions. We highlight our main experimental additions here and then address individual reviewer concerns in each reviewer response:

- **Meta-RL baseline (Reviewer 8qBK)**: We conduct a comparison with a meta-RL baseline, RL2 [1] that performs recurrent meta-learning. We train RL2 on a set of training tasks and evaluate its adaptation performance on out-of-distribution tasks to demonstrate the transfer behavior under distribution shift. Since RL2 adapts in a very short context (2 episodes), we plot the results at the end of adaptation as horizontal lines with error bars. From Fig. 1, we see that even with privileged information during training, RL2 does not handle distribution shift during testing and often performs poorly on test-time tasks. We note that the good performance on Door Open results from the goals being largely in the same direction. So even with the task separation, RL2 still bootstraps behavior from the training tasks.
- **MOPO baseline (Reviewer BJ94)**: We conduct a comparison with an offline model-based RL baseline as requested, MOPO [2]. While this is not quite in the same problem setting, we adapted using the same assumptions as in CQL, running a goal conditioned variant of this method. As shown in Fig 1 in the rebuttal PDF, we found that RaMP outperforms MOPO across all Metaworld tasks.
- **IQL baseline (Reviewer gqpD)**: We ran an IQL baseline on Metaworld domains as shown in Fig 1. We found that RaMP significantly outperformed IQL in both efficiency and asymptotic performance.
- **D4RL experiments (Reviewer gqpD)**: We conducted experiments using the D4RL dataset on 2 maze environments - U Maze and Medium Maze name them. It is important to note that we are not in the standard offline RL setting, since we do not assume known reward on the offline dataset, only some at adaptation time. Moreover, the focus of RaMP is really on the transfer performance, so the standard D4RL comparisons are not quite comparing apples to apples. Our results show that RaMP is applicable to standard offline datasets in D4RL.
- **Ablations wrt added implementation details (Reviewer BJ94)**: We conducted an ablation, removing each of the added components in the implementation details outlined in Section B. We find that each of the design contributes to the overall performance of our method.

We provide additional clarifications, explanations and discussion in the per-reviewer responses.

[1] Yan Duan, John Schulman, Xi Chen, Peter L. Bartlett, Ilya Sutskever, Pieter Abbeel. RL2: Fast Reinforcement Learning via Slow Reinforcement Learning. ICLR 2017.

[2] Tianhe Yu, Garrett Thomas, Lantao Yu, Stefano Ermon, James Zou, Sergey Levine, Chelsea Finn, Tengyu Ma. MOPO: Model-based Offline Policy Optimization. NeurIPS 2020.

---

### Decision · Program_Chairs · 2023-09-21

**Decision:**

Accept (poster)

**Comment:**

There were some mixed opinions on the papers, but overall the positive aspects outweigh the negative. One point of difficulty in assessing the paper was that the problem setting did not allow for easy direct comparisons to algorithms that superficially appear to be comparable. The authors did a reasonable job in articulating this in the rebuttal and with follow-up experiments. They should take this point of confusion into account, as well as other critical points made by reviewers, when preparing the camera ready paper.